# Reliability Scaling Laws for Quantized Large Language Models

**Sirine Ayadi**\*                                                                                                    *sirine.ayadi@tum.de*
*School of Computation, Information and Technology, Technical University of Munich*
*Munich Data Science Institute*

**Sándor Daróczi**\*                                                                                               *sandor.daroczi@tum.de*
*School of Computation, Information and Technology, Technical University of Munich*
*Munich Data Science Institute*

**Stephan Günnemann**                                                                                       *s.guennemann@tum.de*
*School of Computation, Information and Technology, Technical University of Munich*
*Munich Data Science Institute*
*Pruna AI*

**Bertrand Charpentier**                                                                              *bertrand.charpentier@pruna.ai*
*Pruna AI*

**Reviewed on OpenReview:** *https://openreview.net/forum?id=UUBijehMQO*

## Abstract

*Quantization* is a powerful strategy to build capable and resource-efficient large language models (LLMs) by reducing the bitwidth of the parameters. While quantized LLMs achieve state-of-the-art performance on unperturbed inputs using standard predictive metrics, their performance on perturbed inputs, measured using reliability metrics, remains underexplored, despite its importance for reliable deployment. To address this gap, we first conduct a comprehensive reliability evaluation of quantized LLMs consisting of three key components: **(1)** *Uncertainty:* We assess the trustworthiness of LLMs quantized to 2, 3, 4, and 8 bits using six different quantization methods, employing established uncertainty metrics. **(2)** *Calibration:* We assess how well-calibrated the uncertainty estimates of quantized models are across model scales and bit precisions. **(3)** *Robustness:* We design character-level and word-level input perturbations to evaluate the reliability of quantized models under semantically-preserving variations in the inputs that arise in real-world applications. Second, we characterize how reliability scales with the total number of model bits. Our study reveals that while the performance scales monotonically with the total number of bits, the reliability scalings are nonlinear. A reliability peak occurs for 4-bit quantized models, indicating that quantizing moderately sized models offers the best reliability-efficiency trade-off. Additionally, our empirical findings reveal that quantization enhances the robustness of LLMs to natural input perturbations.

## 1 Introduction

Large language models (LLMs) (Touvron et al., 2023; Team et al., 2024; Wei et al., 2022) have reshaped the field of Natural Language Processing (NLP) with their remarkable performance across a range of complex benchmarks. Recent advances in LLMs are based on the principle that model performance improves predictably with increased model size and training data, following well-characterized scaling laws (Kaplan et al., 2020). However, their large size and high computational needs pose significant challenges for practical

---

\*Equal contribution.

use, especially in resource-limited settings. This has spurred interest in model compression to reduce inference latency and memory requirements, including quantization (Dettmers et al., 2022; Lin et al., 2024a; Frantar et al., 2022), pruning (Sun et al., 2023; Frantar & Alistarh, 2023), and knowledge distillation (Gu et al., 2023; Liang et al., 2023).

Despite rapid progress in LLM quantization, their evaluation has predominantly focused on benign task performance, emphasizing that compressed models should retain the accuracy of the base model on downstream tasks (Lin et al., 2024a; Frantar et al., 2022; Sun et al., 2023; Frantar et al., 2025). While useful, these evaluations ignore critical reliability aspects, notably uncertainty and robustness, which are essential for a safe and trustworthy deployment. On the one hand, uncertainty quantification (Kadavath et al., 2022) and calibration (Krishnan et al., 2024) have gained significant traction to assess the trustworthiness of responses generated by LLMs. However, the impact of model quantization on these critical dimensions remains underexplored. On the other hand, quantized models deployed in practice may encounter minor input noise, such as typos or grammatical errors, while the semantics of the original sentence are preserved (Moradi & Samwald, 2021). Yet, commonly used benchmarks (Joshi et al., 2017; Reddy et al., 2019) for evaluating the capabilities of quantized models fail to capture such perturbations, limiting our understanding of model robustness in real-world applications.

To address these gaps, we conduct a comprehensive evaluation of quantized LLMs across multiple **reliability** dimensions, including *uncertainty*, and *robustness* to semantically-preserving input perturbations. To better understand how many bits should be used to improve the reliability-efficiency trade-off, we leverage **bit-level scaling laws**, which reveal underlying trends that extend beyond individual data points. Consider two models: One model trained from scratch with 4 billion parameters at 16-bit precision, and another model with 16 billion parameters quantized to 4-bit precision. While both have the same total number of bits, their behavior can differ significantly (Dettmers & Zettlemoyer, 2023). Prior work (Xu et al., 2024b; Dutta et al., 2024; Lin et al., 2024a; Dettmers & Zettlemoyer, 2023) mainly focus on assessing the effectiveness of the quantization methods by solely relying on the downstream task performance. In particular, Dettmers & Zettlemoyer (2023) explore bit-level scaling trends for standard task performance on different benchmarks, showing that accuracy typically improves with the total number of bits. However, we find that higher accuracy does not necessarily correspond to higher reliability.

As shown in fig. 1, accuracy scales monotonically with the total number of model bits. In contrast, reliability follows a non-monotonic trend: reliability peaks for moderately-sized models quantized to 4 bits, suggesting that an optimal trade-off can be achieved without resorting to either high-precision quantization or the largest model.

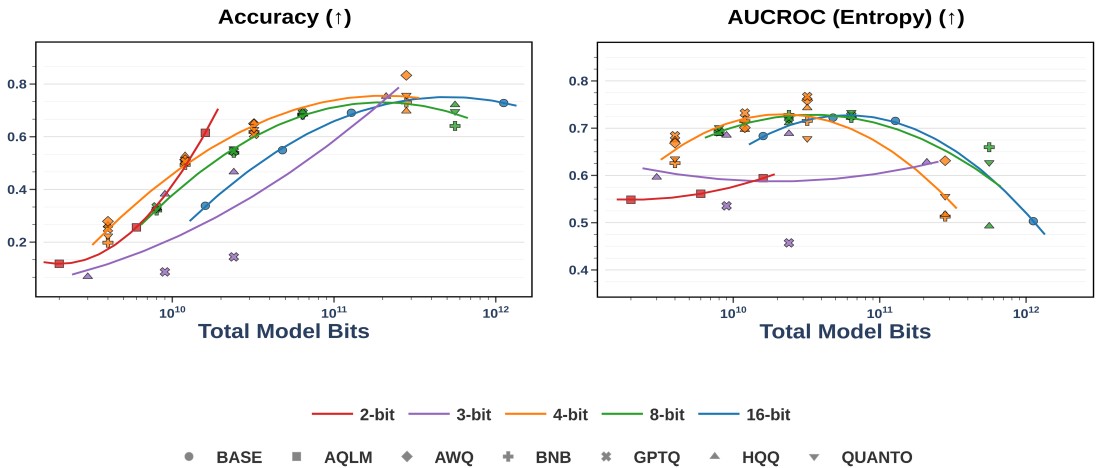

Figure 1: Bit-level scaling trends of the accuracy and AUCROC(Entropy) on TriviaQA. We use four base models (blue): LLaMA-3.2-1B, LLaMa-3.2-3B, LLaMa-3-8B, and LLaMa-3-70B, and all available corresponding quantized variants using six quantization methods and different bitwidths.

We highlight our main contributions in the following:

- We introduce a reliability evaluation framework for full-precision and quantized LLMs, consisting of three key components: (i) predictive uncertainty, (ii) calibration, and (iii) robustness to semantically-preserving perturbations. To this end, we implement 15 natural character-level and word-level perturbations that commonly arise in typed digital communication.

- We characterize reliability bit-level scaling trends to examine the precision that maximizes reliability for a fixed budget of total model bits. We conduct the first comprehensive reliability evaluation of six state-of-the-art quantization techniques across models with 1 billion to 70 billion parameters. We examine whether reliability scaling trends are consistent across different model architectures and benchmarks. We find that 4-bit precision offers the best trade-off between reliability and efficiency across tasks, model families, and quantization methods.

- We further investigate why 4-bit precision yields favorable performance, and examine whether the observed scaling trends persist under different LLM pruning strategies.

## 2 Related Work

**Scaling Laws for LLMs**  This work builds on established scaling laws that characterize how the training budget affects the performance of language models (Kaplan et al., 2020; Hernandez et al., 2021; Sorscher et al., 2022; Kumar et al., 2024). Kaplan et al. (2020) demonstrate that the model loss follows a power law with the number of model parameters and tokens. Building on this, Hoffmann et al. (2022) argue that both model size and the number of training tokens should be scaled equally to achieve optimal compute training. Other works have shifted focus to the test-time budget, examining how quantization can impact the performance (Dettmers & Zettlemoyer, 2023; Kumar et al., 2024; Frantar et al., 2025). In this work, our goal is to understand the impact of the compression budget, defined as the total number of model bits, on downstream performance. Closest to our work are the bit-level scaling laws for zero-shot performance of quantized LLMs proposed in Dettmers & Zettlemoyer (2023), where the goal is to determine the precision that maximizes the accuracy. However, Dettmers & Zettlemoyer (2023) solely focus on scaling trends of downstream task performance, and do not account for critical reliability dimensions necessary for safe deployment.

**Reliability Evaluation of LLMs**  With the increasing interest in LLMs, understanding uncertainty and calibration is crucial. Various uncertainty quantification methods have emerged to determine the trustworthiness of the generated responses by LLMs (Kuhn et al., 2023; Ye et al., 2024). Kuhn et al. (2023) introduce semantic entropy, which considers linguistic invariants arising from shared meanings. Malinin & Gales (2020) introduce predictive entropy, which quantifies the uncertainty of auto-regressive models based on their own outputs. Recently, Ye et al. (2024) employed conformal prediction to quantify the uncertainty of LLMs in NLP tasks, including question answering and reading comprehension. Calibrating the uncertainty estimates is essential to evaluating the reliability of LLMs. A well-calibrated model correlates low uncertainty with accurate responses and high uncertainty with likely incorrect responses. This is typically evaluated using the Expected Calibration Error (ECE) (Jiang et al., 2021) and the Brier Score (Brier, 1950). In generative tasks, defining calibration is often challenging (Kapoor et al., 2024), especially for variable-length response sequences.

While several studies have explored different reliability dimensions for LLMs, these aspects remain underexplored for quantized models. Most prior works on model compression, such as Lin et al. (2024a); Frantar et al. (2022); Frantar & Alistarh (2023); Team (2023); Ashkboos et al. (2024), primarily evaluate the effectiveness of compression methods through standard performance metrics, including perplexity and accuracy on various benchmarks. However, these benchmarks fail to capture critical aspects of model reliability. Recently, Hong et al. (2024) present a comprehensive evaluation of compressed LLMs across various trustworthiness dimensions including stereotype, toxicity, privacy, fairness, ethics, adversarial robustness, and out-of-distribution robustness. Their primary question of interest is how to construct trustworthy 7B models, either by pre-training from scratch or by compressing a larger pre-trained 13B model. (Hong et al., 2024) reveal that quantization offers a more favorable trade-off between efficiency and trustworthiness than pruning.

## 3 Reliability Evaluation Framework

Consider a pre-trained autoregressive language model $P_\theta\left(\mathbf{y} \mid \mathbf{x}\right)$ parameterized by $\theta$, where $\mathbf{x}$ is an input prompt, and $\mathbf{y}$ is the generated sequence. We adapt this model to downstream conditional text generation tasks, such as question answering. Each task is represented by a dataset of context-target pairs, $\mathcal{D} = \{(\mathbf{x}_i, \mathbf{y}_i^*)\}_{i=1}^N$, where both $\mathbf{x}_i$ and $\mathbf{y}_i^*$ are sequences of tokens. Given an input prompt $\mathbf{x}$, the model generates answer tokens sequentially as follows:

$$P_\theta\left(\mathbf{y} \mid \mathbf{x}\right) = \prod_{t=1}^T P_\theta\left(\boldsymbol{y}_t \mid \boldsymbol{y}_1, \ldots, \boldsymbol{y}_{t-1}, \mathbf{x}\right), \tag{1}$$

where $\mathbf{y} = (\boldsymbol{y}_1, \ldots, \boldsymbol{y}_T)$ is the final generated sequence consisting of $T$ tokens. We denote the model's predicted token distribution by $P_\theta\left(\boldsymbol{y} \mid \boldsymbol{y}_1, \ldots, \boldsymbol{y}_{t-1}, \mathbf{x}\right)$, where $\boldsymbol{y} \in \mathbb{R}^{|\mathcal{V}|}$, and $\mathcal{V}$ is the vocabulary. At each decoding step $t$, the model samples a token $\boldsymbol{y}_t \sim P_\theta(. \mid \boldsymbol{y}_{1:t-1}, \mathbf{x})$.

To study the bit-level scaling, we consider quantization, which is a widely used compression technique that reduces the number of bits in the parameters of a model with minimal loss in inference performance (Dettmers et al., 2022; Lin et al., 2024a; Frantar et al., 2022; Team, 2023; Badri & Shaji, 2023).

### 3.1 Reliability Evaluation

In this paper, our goal is to evaluate various aspects of *reliability* for quantized LLMs in a question answering setting. Given an input prompt $\mathbf{x}$, $P_\theta(\mathbf{y} \mid \mathbf{x})$ spans a probability distribution of all possible output sequences. Although we are generally interested in the output distribution $P_\theta(\mathbf{y} \mid \mathbf{x})$, in practice, we cannot directly access this distribution since the number of possible output sequences $|\mathcal{V}|^T$ is large. Therefore, we first define our reliability metrics at the *token-level*, as we have direct access to the token distributions $P_\theta(\boldsymbol{y} \mid \boldsymbol{y}_{1:t-1}, \mathbf{x})$ at every step $t$, and aggregate these measures to obtain *sequence-level* metrics. Let $M_t$ denote a token-level metric at step $t$, the sequence-level metric is computed as the average across all time steps:

$$M_{\mathrm{seq}} = \frac{1}{T} \sum_{t=1}^T M_t. \tag{2}$$

Relying on a single generated sequence $\mathbf{y}$ given an input prompt $\mathbf{x}$ can lead to an incomplete assessment in a natural language generation (NLG) setting. To provide a more robust and representative estimate of the model's reliability, we sample multiple sequences per input prompt and report averaged results across these samples. In the following, we outline the different reliability metrics measured on the token level. To measure the model's uncertainty in the predicted tokens, we compute the *token-level entropy*, which corresponds to the Shannon entropy of the token distribution (Fomicheva et al., 2020; Malinin & Gales, 2020):

$$H_t = -\sum_{k=1}^{|\mathcal{V}|} P_\theta\left(\boldsymbol{y}_k \mid \boldsymbol{y}_{1:t-1}, \mathbf{x}\right) \log P_\theta\left(\boldsymbol{y}_k \mid \boldsymbol{y}_{1:t-1}, \mathbf{x}\right), \tag{3}$$

measured across the entire vocabulary $\mathcal{V}$. Higher entropy corresponds to increased uncertainty in the predicted next-token distribution, while lower entropy corresponds to increased confidence. To quantify the model's uncertainty in the correct next-token, we compute the *token-level log-likelihood* of the ground-truth token $\boldsymbol{y}_t^*$ as follows:

$$C_t^* = \log P_\theta\left(\boldsymbol{y}_t^* \mid \boldsymbol{y}_{1:t-1}^*, \mathbf{x}\right). \tag{4}$$

To assess how well-calibrated the model's predicted token distributions are, we adopt the *Brier Score* (Brier, 1950), and define the token-level calibration as:

$$\mathrm{Brier}_t = \sum_{k=1}^{|\mathcal{V}|} \left(P_\theta\left(\boldsymbol{y}_k \mid \boldsymbol{y}_{1:t-1}, \mathbf{x}\right) - \boldsymbol{y}_k^*\right)^2, \tag{5}$$

which corresponds to the squared Euclidean distance between the predicted token distribution $P_\theta\left(\boldsymbol{y} \mid \boldsymbol{y}_{1:t-1}, \mathbf{x}\right)$ and the one-hot encoded reference token.

| Original prompt: | What is the capital of France? |
|---|---|
| **Character-Level Perturbations** | |
| 1. Insertion | What is the capital of **r**France? |
| 2. Deletion | What is the cap**_**tal of France? |
| 3. Replacement | What is the capital of Fr**e**nce? |
| 4. Swapping | What is t**eh** capital of France? |
| 5. Repetition | Wha**tt** is the capital of France? |
| 6. Leetspeak | What is the c**@**pital of France? |
| 7. Noise | What is the capital of Fra**!**nce? |
| 8. Case Change | What is the capital o**F** France? |
| 9. Emoji | What is the capital of France? 😊 |
| **Word-Level Perturbations** | |
| 10. Insertion | What is the capital **city** of France? |
| 11. Deletion | What **_** the capital of France? |
| 12. Swapping | **is What** the capital of France? |
| 13. Repetition | What **What** is the capital of France? |
| 14. Slang | What is the capital of France **rofl**? |
| 15. Translation | What is the capital of **法国**? |

Figure 2: Overview of our perturbations. Illustrated is an example where perturbations with intensity level 1 are applied to a standard question prompt.

## 3.2 Robustness to Input Perturbations

In this work, we focus on natural, semantically-preserving perturbations commonly encountered in typed digital communication, such as messaging or chat-based applications. We introduce character-level and word-level perturbations to the input prompt $\mathbf{x}$. We provide an overview of all input perturbations in fig. 2. We design natural perturbations that are increasingly present in online communication (Suhardianto et al., 2019; Moradi & Samwald, 2021; Hand et al., 2022; Ackerman et al., 2024), but remain underexplored in the robustness evaluation of base and quantized LLMs. We implement the following perturbations:

- *Emoji:* Emojis are increasingly integrated into the written language in interpersonal communication. Current digital communication is more likely to include emojis alongside text rather than traditional emoticons (Hand et al., 2022). To evaluate the model's robustness to emojis, we perturb the input prompt by randomly inserting emoji characters.

- *Replacement*: We substitute characters with adjacent keyboard alternatives to simulate realistic typos, using a keyboard layout mapping that maintains case sensitivity (Ackerman et al., 2024).

- *Slang:* Slang expressions are commonly used to communicate informally in online dialogues (Suhardianto et al., 2019). To test the model's sensitivity to informal internet language, we introduce perturbations by randomly inserting slang expressions such as "lol", "rofl", and "IMO".

- *Insertion:* At the word level, we add contextually relevant words using a language model (Liu et al., 2019) to predict insertions while preserving text coherence. At the character level, we randomly insert characters.

- *Leetspeak:* We substitute characters with visually similar numerals or symbols (for example, we replace 'b' with $'6'$). We provide additional examples in table 4.

- *Translation:* To simulate the linguistic behavior of individuals with different languages, we randomly translate words to six common languages. Contrary to the semantic-level translation perturbation introduced in Zhu et al. (2024), we do not translate phrases back to English.

- *Deletion:* In the process of deletion, we only remove filler words (e.g., "and", "to", or "actually") that are not crucial for preserving the semantics. For character deletion, we randomly delete characters

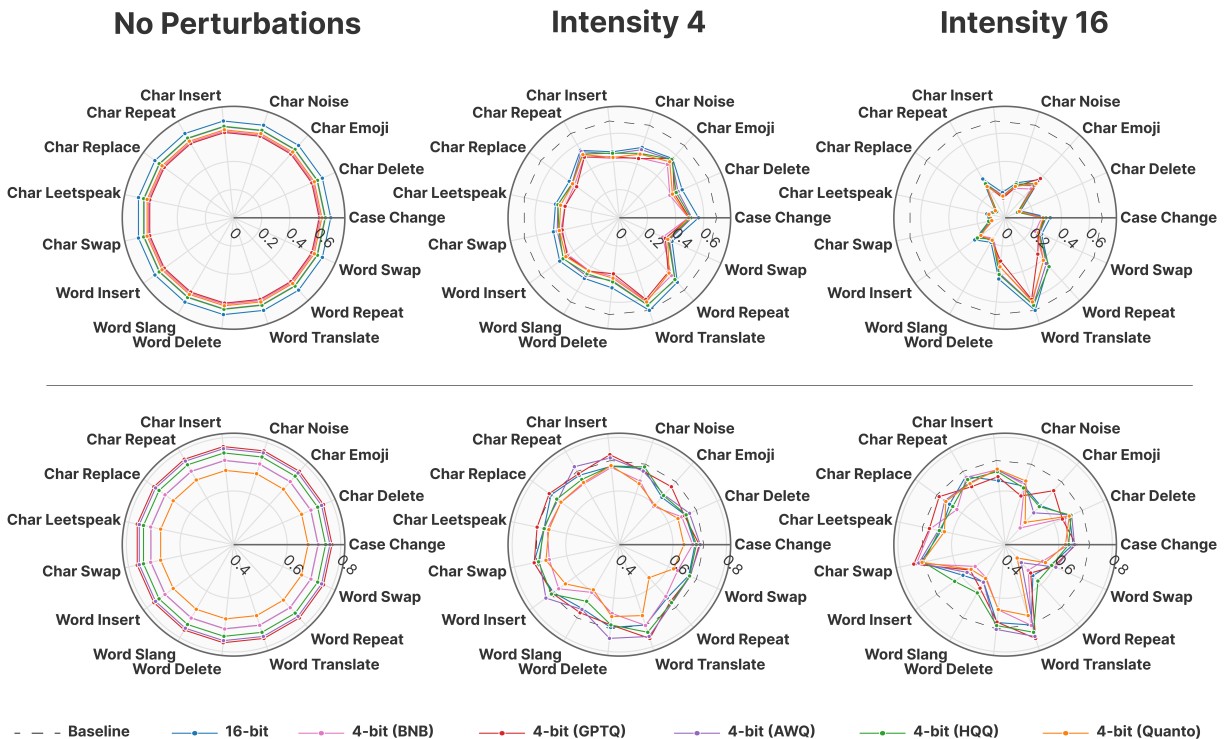

Figure 3: Radar plots of the accuracy (**Top**) and AUCROC (Entropy) (**bottom**) across all 15 character-level and word-level perturbations for two intensities. We evaluate the base LLaMa-3-8B model and five 4-bit quantization methods. Quantized models can provide more reliable uncertainty estimates under natural perturbations compared to their base counterparts, while maintaining a close performance.

from the sequence. This perturbation simulates typos that are commonly observed in informal or fast-paced writing (Flor et al., 2015).

We further design standard character-level and word-level perturbations such as swapping, repetition, and case change (Moradi & Samwald, 2021; Ackerman et al., 2024). Ackerman et al. (2024) test many of the character-level perturbations in adversarial attack scenarios. However, in our work, we focus on a set of non-adversarial perturbations that can be used to assess the robustness of LLMs. Additional details on all perturbations are provided in section B. We provide radar plots of the accuracy and AUCROC (Entropy) on the unperturbed and perturbed TriviaQA dataset across all perturbations in fig. 3. We use the full precision LLaMA-3-8B model and five quantized models. We show that 4-bit quantization does not degrade the performance under perturbations. For the reliability evaluation, we report the AUCROC (Entropy) scores, and find that quantization methods, including GPTQ (Frantar et al., 2022), AWQ (Lin et al., 2024a), and HQQ (Badri & Shaji, 2023), improve reliability under natural perturbations.

## 4 Bit-Level Scaling Trends

Consider two models: *(i)* a model with $K_1$ parameters in $P_1$-bit precision, and *(ii)* a model with $K_2$ parameters in $P_2$-bit precision, such that $K_1P_1 = K_2P_2$. Although both models have equal total bit budgets, their reliability characteristics can differ significantly. In this paper, our goal is to characterize how reliability scales with the total number of bits. Using total model bits as a unit for comparison enables a fair model comparison across different parameter counts and bitwidths. We note that we do not propose a formal predictive law in the classical sense; i.e., a closed-form equation that predicts outcomes a priori before evaluation. In prior works (Dettmers & Zettlemoyer, 2023; Frantar et al., 2025), scaling laws often refer to empirical regularities observed in model behavior, as a function of model size, compute, or dataset size. These laws are typically

fit to the observed data and used to characterize trends. We adopt the same empirical methodology: we empirically characterize consistent trends in reliability behavior as the total number of bits $B$ scales, defined as the product of the number of model parameters and the weight precision (in bits). Consider a performance metric $\mathcal{L}$ modeled using a *log quadratic scaling law* as follows:

$$\mathcal{L}(B) = a(\log(B))^2 + b\log(B) + c\,, \tag{6}$$

where $a, b, c \in \mathbb{R}$ are fitted coefficients. This formulation captures nonlinear trends in model performance as the model capacity increases, and generally provides a good fit, as shown in section 5.

**Why is total model bits a reasonable axis of comparison?** Total model bits provide a unified basis for systematically studying trade-offs between model scale and quantization bitwidth. It is both convenient and easy to measure, while also correlating well with several practical efficiency metrics. In particular, total model bits directly reflect the storage memory and scale linearly with the inference memory. We further discuss in section C how it relates to inference latency and throughput. table 1 provides a quantitative comparison of efficiency metrics for Qwen models (Yang et al., 2025) with comparable total bit budgets.

Table 1: We report the total model bits (in GB), the inference memory (in GB), the per-token latency (ms), corresponding to the time spent to generate each token in the output, and the number of decoded tokens per second. We evaluate on a single A100 GPU.

| Model | Bitwidth | Total Model Bits | Inference Memory | Per-Token Latency | Decoded tokens per second |
|---|---|---|---|---|---|
| Qwen3-4B | 16-bit | 8.217 | 8.409 | 4.103 | 266.35 |
| Qwen3-4B | 8-bit | 4.108 | 4.300 | 2.510 | 462.66 |
| Qwen3-8B | 16-bit | 15.256 | 15.476 | 9.884 | 110.30 |
| Qwen3-8B | 8-bit | 7.628 | 7.848 | 5.806 | 200.50 |
| Qwen3-8B | 4-bit | 3.814 | 4.034 | 3.766 | 339.16 |
| Qwen3-32B | 4-bit | 15.250 | 15.478 | 8.719 | 150.29 |

## 5 Reliability Assessment of Quantized LLMs

We conduct a comprehensive evaluation of the reliability of quantized LLMs both on unperturbed and perturbed input prompts, across different model scales, families, and precisions. We address the following questions: How does the downstream task performance scale as a function of total model bits? Do the reliability scalings exhibit different trends? What is the optimal precision for trading off reliability and efficiency? Additionally, we conduct ablation studies to investigate whether these scaling behaviors persist across model families and benchmarks.

### 5.1 Experimental Setting

**Datasets:** We consider commonly used benchmarks for evaluating the uncertainty quantification in LLMs across NLG tasks (Lin et al., 2023; Kuhn et al., 2023): **TriviaQA** (Joshi et al., 2017) and **CoQA** (Reddy et al., 2019). TriviaQA (Joshi et al., 2017) evaluates factual knowledge retrieval with question-answer pairs sourced from trivia competitions. It spans diverse knowledge domains, including history, literature, science, geography, and popular culture. CoQA (Reddy et al., 2019) (Conversational Question Answering) evaluates contextual understanding through multi-turn conversational exchanges. The dataset consists of conversations from seven diverse domains, including literature, news, and science. Additionally, we evaluate on **CommonsenseQA** (Talmor et al., 2018), a multiple-choice question-answering dataset designed to assess commonsense knowledge and reasoning capabilities, where the goal is to provide a single answer from five possible answer choices. To assess the language modeling capabilities, we evaluate on **WikiText-2** (Merity et al., 2016), **PTB** (Marcus et al., 1993), and **C4** (Raffel et al., 2020). Additional details are presented in section A.2. We extend the evaluation to additional benchmarks to assess the emergent abilities of base and quantized models on in-context learning and instruction-following tasks, as well as other NLP tasks such as reasoning and comprehension in section D. A summary of all evaluated benchmarks is provided in table 3.

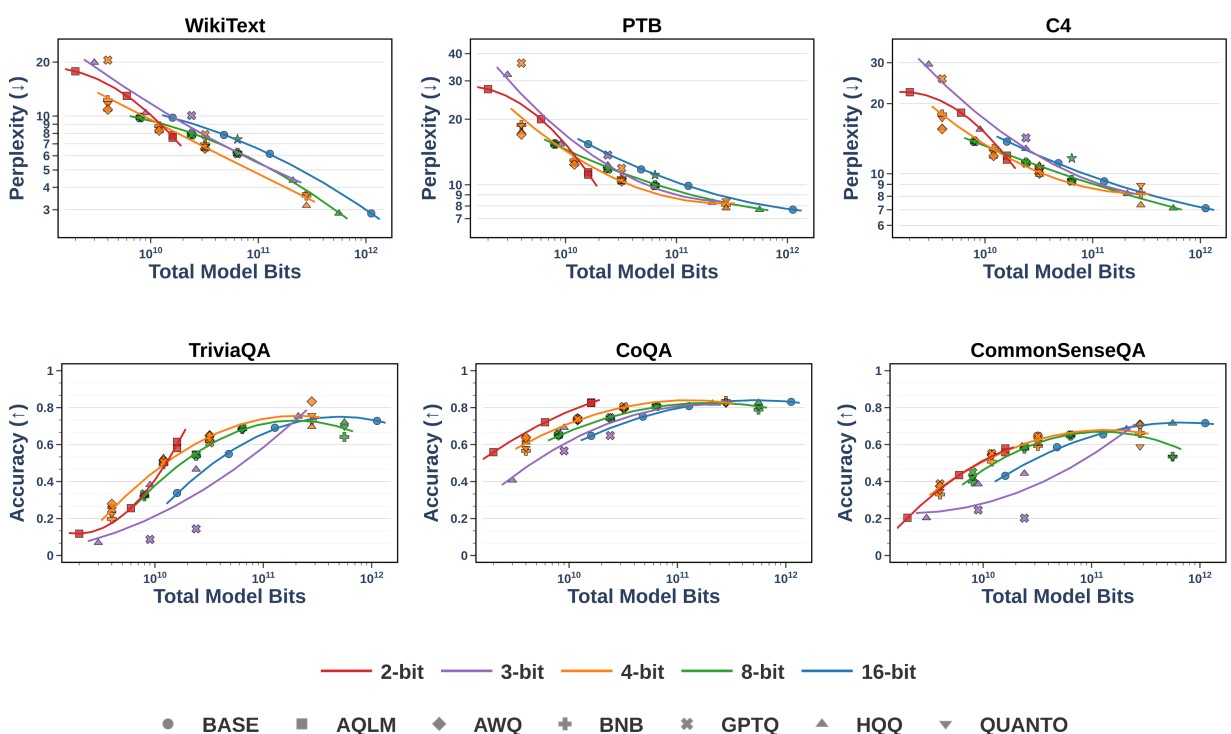

Figure 4: Scalings of the perplexity (**top**) and accuracy (**bottom**) for all quantized models and their corresponding full-precision models. The performance steadily improves with the total number of model bits. 4-bit models offer the best performance-efficiency trade-off given a fixed number of total model bits.

**Base and quantized LLMs.** We consider four base pre-trained models (Grattafiori et al., 2024), including models from the LLaMA-3.2 family: LLaMA-3.2-1B and LLaMA-3.2-3B, and models from the LLaMA-3 family: LLaMA-3-8B and LLaMA-3-70B. For the quantization, we consider six state-of-the-art quantization methods across various precisions, ranging from 2 bits to 8 bits. These approaches include BitsandBytes (Dettmers et al., 2022), AWQ (Lin et al., 2024a), GPTQ (Frantar et al., 2022), HQQ (Badri & Shaji, 2023; 2024), Quanto (Team, 2023), and AQLM (Egiazarian et al., 2024; Malinovskii et al., 2024). For the 2-bit quantization using AQLM, we adopt the AQLM-PV variant, which is a quantized model with additional fine-tuning for improved performance. A complete list of all base models, quantized models, and corresponding precision levels is presented in section A.1. To better capture performance trends, we fit log-quadratic functions as outlined in section 4.

## 5.2 Bit-Level Scaling Trends

**Downstream task performance scalings.** We evaluate the downstream performance of base and quantized LLMs on *unperturbed* (i.e., original) datasets. In fig. 4, we present the scaling behavior of the perplexity and the accuracy across different quantization levels and model sizes. First, we observe that the downstream task performance improves with increased total model bits. Second, we find that reducing the precision from 16 to 4 bits, for a given total bit budget, consistently improves both the perplexity and the accuracy across all evaluated datasets. However, this trend is reversed if we further decrease the precision to 3 bits and 2 bits, aligning with previous observations in Hong et al. (2024); Dettmers & Zettlemoyer (2023). We note that the 2-bit quantization using AQLM (Egiazarian et al., 2024; Malinovskii et al., 2024) achieves a relatively good performance in the low-bit regime on CoQA and CommonsenseQA. For the 3-bit quantization, HQQ (Badri & Shaji, 2023) achieves the best performance. For the 4-bit setting, both HQQ (Badri & Shaji, 2023) and AWQ (Lin et al., 2024a) offer the best performance-efficiency trade-off.

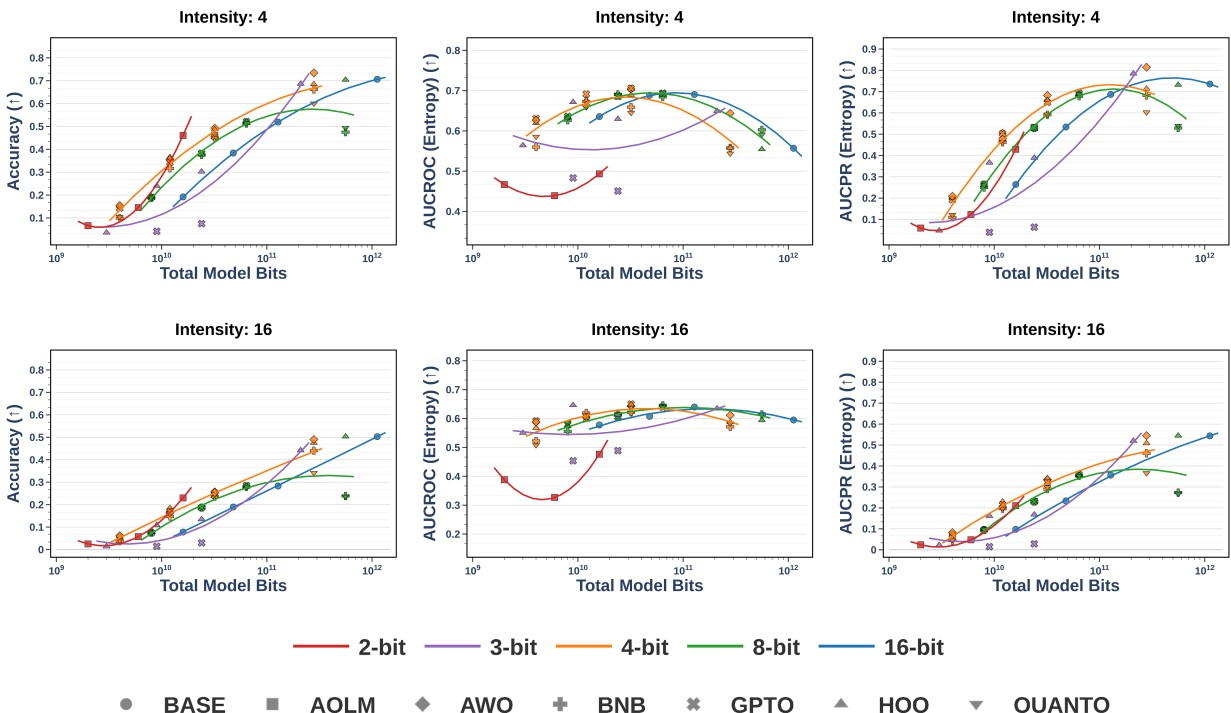

Figure 5: Bit-level scalings for all evaluated quantized models and their corresponding full-precision models on the perturbed TriviaQA dataset, averaged over all 15 perturbations.

**Reliability scaling trends.** To characterize the reliability of base and quantized models, we present the scaling behavior of the accuracy and the reliability metrics on the perturbed TriviaQA dataset in fig. 5. We average the metrics across all 15 perturbations and provide a qualitative comparison for two different perturbation intensities. On the one hand, accuracy increases under both unperturbed and perturbed prompts with the total number of model bits. Additionally, 4-bit models offer the strongest robust accuracy for a fixed bit budget. We note that the 3-bit HQQ quantization of the LLaMA-3-70B model achieves an accuracy comparable to its 4-bit counterpart. On the other hand, bit-level reliability scalings are not linear. More specifically, larger models exhibit lower AUCROC (Entropy), indicating reduced reliability under perturbations. Extreme 2-bit quantization methods exhibit different scaling trends than moderate precisions. While 4-bit, 8-bit, and 16-bit models have worse uncertainty estimates and calibration in the large-bit regime, the reliability of 2-bit models improves with model scale. Additional results are provided in section E.

**Why does 4-bit quantization offer a favorable reliability-efficiency trade-off?** The goal of model quantization is to create a more efficient model from a full-precision base model, while maintaining as close a distance to its full-precision counterpart. While both perplexity and accuracy metrics are essential for evaluating the generalization capabilities of quantized models, they fail to capture the shifts in model behavior that can occur post-compression. To quantify this behavioral shift, we measure the Kullback-Leibler Divergence between the token distributions predicted by the base model, $P_{\theta_B}(\boldsymbol{x}_t \mid \boldsymbol{x}_{<t})$, and those predicted by the compressed model, $P_{\theta_C}(\boldsymbol{x}_t \mid \boldsymbol{x}_{<t})$. Let $\mathbf{x} = (\boldsymbol{x}_1, \ldots, \boldsymbol{x}_T) \sim \mathcal{D}_{\text{test}}$ be a sequence sampled from a test dataset, and $T$ defines the number of tokens in the

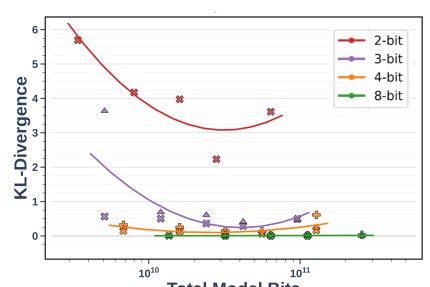

Figure 6: Scaling trends of the KL-Divergence.

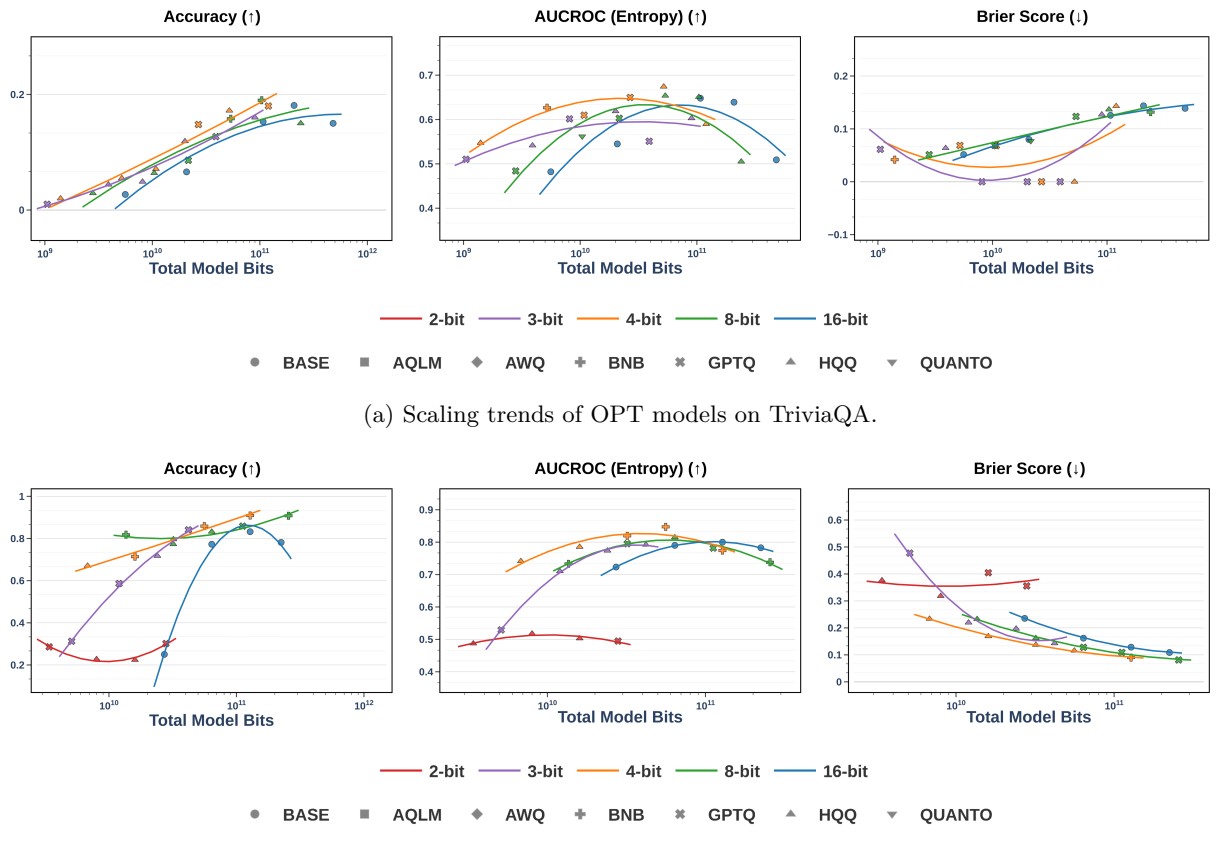

(a) Scaling trends of OPT models on TriviaQA.

(b) Scaling trends of Qwen3 models on CEval.

Figure 7: Scaling trends of OPT and Qwen3 models across different benchmarks.

sequence. The behavioral shift is measured as follows:

$$\frac{1}{|\mathcal{D}_{\text{test}}|} \sum_{\mathbf{x} \in \mathcal{D}_{\text{test}}} \frac{1}{T} \sum_{t=1}^{T} \text{KL}\left(P_{\theta_B}(\boldsymbol{x}_t \mid \boldsymbol{x}_{<t}) \parallel P_{\theta_C}(\boldsymbol{x}_t \mid \boldsymbol{x}_{<t})\right) . \tag{7}$$

In fig. 6, we provide the bit-level scaling trends of the KL-Divergence. We consider 3 LLaMA models with 1B, 3B, and 8B parameters, and quantize them to different bitwidths. First, we observe that 2-bit quantized models exhibit a substantial behavioral shift relative to their full-precision counterparts, which contributes to their consistently poor performance. For 3-bit quantization, the KL divergence remains noticeably larger than that of 4-bit models. However, as model size increases from 1B to 8B parameters, the behavioral shift induced by 3-bit quantization diminishes. This reduction helps explain the improved downstream performance of larger 3-bit models (see fig. 4). In contrast, 4-bit quantization maintains a small KL divergence across model scales. We believe that this small behavioral shift allows 4-bit models to retain strong generalization capabilities while still achieving a substantial reduction in total model bits.

While the KL divergence of 8-bit models is better than that of 4-bit models, the key additional factor favoring 4-bit quantized models is their higher model capacity within a fixed total-bit budget. A 4-bit model matched to an 8-bit model in total bits has roughly twice the number of parameters. Our results suggest that, in this regime, the benefit of increased model capacity can outweigh the benefit of higher weight precision: 4-bit models remain sufficiently close to their full-precision counterparts while benefiting from a larger parameter count. Below 4 bits, this trade-off changes: the behavioral shift induced by aggressive quantization becomes too large to be compensated by additional parameters.

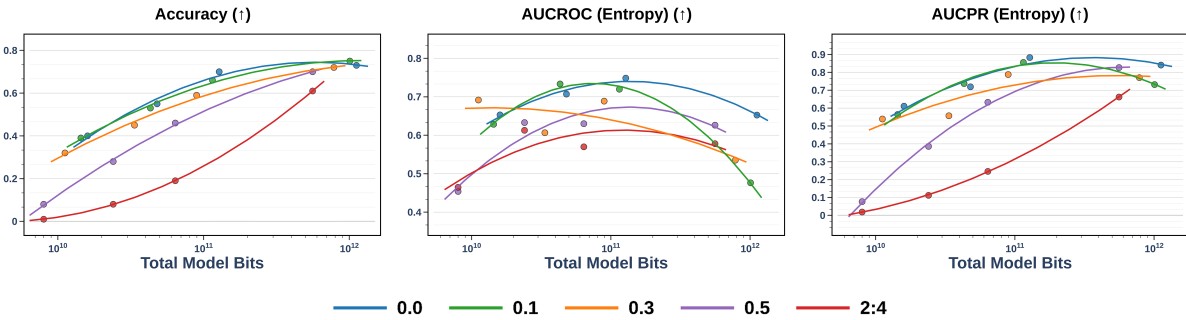

(a) Scaling trends of models pruned using SparseGPT to different sparsities.

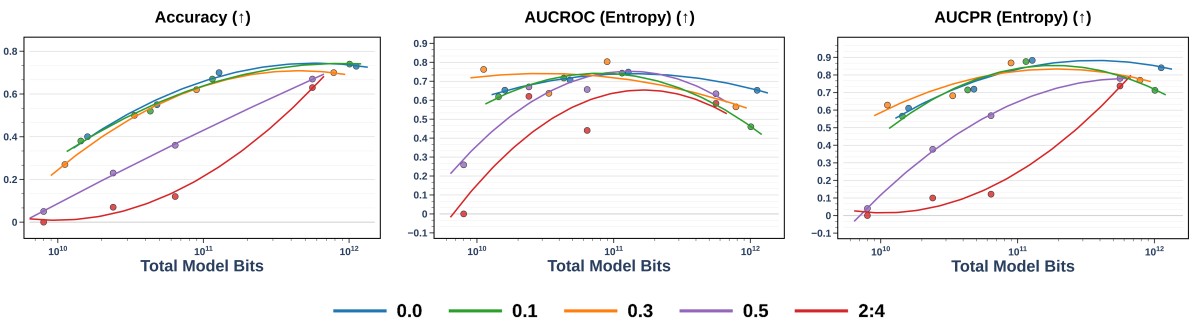

(b) Scaling trends of models pruned using Wanda to different sparsities.

Figure 8: Scaling trends of pruned LLaMA models using SparseGPT and Wanda.

**Are the scaling trends consistent across diverse model architectures?** We extend our analysis beyond the LLaMA-3 family to include models from OPT (Zhang et al., 2022) and Qwen3 (Yang et al., 2025) series. For OPT, we evaluate six base models with parameter counts of 350M, 1.3B, 2.7B, 13B, and 30B, and present qualitative results in fig. 7a. For Qwen3, we consider models of sizes 4B, 8B, 14B, and 32B, and present results in fig. 7b. Across different architectures, we observe a consistent 4-bit sweet spot: 4-bit quantization generally achieves the best trade-off between accuracy, reliability, and efficiency. Moreover, the scaling behavior aligns with our earlier findings. Interestingly, for Qwen models, we find that the extreme 3-bit quantization of moderate-to-large models yields favorable performance. For OPT models, HQQ and GPTQ exhibit the best performance among 4-bit quantization methods, with GPTQ achieving the best results among 3-bit quantizations. For Qwen models, HQQ and BitsandBytes outperform other PTQ approaches in the 4-bit setting, whereas GPTQ and HQQ provide the best performance under 3-bit compression. Additional evaluations on widely used benchmarks for Qwen models are presented in section D.

**How do pruning strategies affect the reliability scaling trends of LLMs?** While our primary focus is to characterize the scaling behavior of PTQ techniques across various model sizes and bitwidths, motivated by their methodological diversity and growing adoption, we also examine pruning, another widely used post-training compression strategy. In particular, we evaluate two state-of-the-art pruning techniques for LLMs: Wanda (Sun et al., 2023) and SparseGPT (Frantar & Alistarh, 2023). We prune four base LLaMA models at sparsity levels of 0.1, 0.3, 0.5, and additionally apply semi-structured 2:4 pruning. The N:M sparsity pattern enforces that only N weights remain non-zero within each block of M consecutive weights. Unlike unstructured pruning, N:M pruning can provide actual hardware acceleration. The corresponding scaling trends for SparseGPT and Wanda are presented in fig. 8a and fig. 8b, respectively, where base models have zero sparsity. We find that as the sparsity increases, both performance and reliability degrade.

While unstructured pruning at low sparsity levels can preserve a performance close to that of their base counterparts, it does not yield substantial gains as observed with 4-bit quantization. This is consistent with prior findings (Harma et al., 2024), which suggest that quantization-induced error is lower than that introduced by sparsification; and therefore, quantized LLMs generally outperform pruned LLMs. While both 50% sparse and semi-structured models exhibit noticeable degradation in reliability, we find that their performance improves significantly as model scale increases. A similar trend is observed for 3-bit quantized models, as shown in fig. 5.

### 5.3  Discussion and limitations

In this paper, we provide the first comprehensive reliability evaluation of base and quantized models by studying the underlying bit-level scaling trends across different model scales and bitwidths. While prior work Jaiswal et al. (2023); Dutta et al. (2024); Dettmers & Zettlemoyer (2023) focus exclusively on zero-shot performance of quantized models, we argue that the downstream task performance is insufficient. Notably, although the performance scales linearly with the total number of bits, with 4-bit quantization generally yielding the highest accuracy, the reliability trends are not linear: A peak occurs for 4-bit models, suggesting that a favorable reliability-efficiency trade-off can be achieved without resorting to the largest model or the highest precision. We further examine the robustness of base and quantized models under natural input perturbations and reveal that 4-bit quantization can enhance the robustness to semantically preserving perturbations that occur in typed digital communication. Our findings are consistent across diverse model architectures and datasets. We further examine the scalings of two pruning methods (Sun et al., 2023; Frantar & Alistarh, 2023), and find that sparsification does not offer significant reliability gains as observed with quantization, aligning with recent findings (Harma et al., 2024).

For future work, exploring how model calibration, multi-shot prompting, and model fine-tuning can affect the reliability scalings of quantized LLMs is crucial. Additionally, an interesting direction is studying which hyperparameters improve the reliability scalings for a specific number of model parameters and bit precision. We focus on post-training quantization approaches as they are widely adopted. Future work can extend this study to other compression approaches, such as Quantization-Aware Training (QAT) techniques, which may reveal different trends. This is particularly interesting given the potential of QAT to improve the performance of extreme quantization (Ma et al., 2024), see section G.

## 6  Conclusion

We present a comprehensive evaluation of quantized LLMs across key reliability dimensions, focusing specifically on uncertainty and robustness to semantically-preserving input perturbations. While prior studies have primarily emphasized evaluating the accuracy of quantized models on standard benchmarks, our findings reveal that this narrow focus overlooks critical safety considerations. By studying reliability scaling trends, we show that reliability does not necessarily scale monotonically with the total number of model bits. An optimal reliability-efficiency trade-off can be achieved without resorting to the highest precision or quantizing the largest base model. We emphasize the potential of quantization to improve model reliability, making it a promising approach for deploying trustworthy LLMs in practical settings.

**Broader Impact Statement**

Our research examines the reliability of quantized large language models across multiple dimensions to support their safe and trustworthy deployment in real-world applications. While quantization methods help reduce the energy and emissions impact of machine learning applications, which is an urgent environmental challenge, they may introduce new challenges, such as decreased robustness to semantically-preserving input perturbations. We highlight the need for more comprehensive evaluations to ensure that quantized models are robust and reliable, thereby meeting the standards required for responsible deployment.

**Acknowledgments**

SA, SG, and BC acknowledge funding from the German Federal Ministry of Research, Technology and Space (BMFTR) under grant agreement No. 01IS24072C (COMFORT). This work was also supported by the DAAD programme Konrad Zuse Schools of Excellence in Artificial Intelligence, sponsored by the Federal Ministry of Education and Research.

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

# A Additional details on the experimental setting

## A.1 Quantization methods

We present the complete list of base models and quantized models used in our evaluations in table 2. We consider different quantization methods, including BNB (Dettmers et al., 2022), AWQ (Lin et al., 2024a), GPTQ (Frantar et al., 2022), HQQ (Badri & Shaji, 2023; 2024), Quanto (Team, 2023), AQLM (Egiazarian et al., 2024; Malinovskii et al., 2024), QuaRot (Ashkboos et al., 2024), and QoQ (Lin et al., 2024b), applied across 4 LLaMA models from the LLaMA-3 family. Not all bitwidth configurations are adopted for every base model due to compatibility constraints and computational limitations. In total, our evaluation encompasses 63 distinct model configurations that we evaluate in the reliability assessment framework. As we only report the best-performing model for each bit width in section 5, both QuaRot and Quanto are not included in the scaling laws figures as they do not outperform other quantization techniques. In addition, we note that we use AQLM-PV for the 2-bit quantization, which corresponds to AQLM quantization with an additional PV tuning step. AQLM-PV provides increased performance improvement compared to AQLM. We run $63 * 3 * (1 + 15 * 2) = 5859$ experiments in total, where 63 is the number of models, 3 is the number of the QA datasets, 15 is the number of evaluations, and 2 corresponds to the two perturbation intensities 4 and 16.

For the quantization using Bitsandbytes (Dettmers et al., 2022), HQQ (Badri & Shaji, 2023), Quanto (Team, 2023) and GPTQ (3-bit), we use the Hugging Face Transformers (Wolf et al., 2019) default implementation, licensed under the Apache-2.0 license. GPTQ is licensed under the Apache-2.0 license, while AWQ is licensed under the MIT license. All AQLM, QuaRot, and QoQ models are loaded directly from HuggingFace, licensed under the Apache License 2.0. The LLaMA-3 and LLaMA-3.2 model families we used in our experiments are licensed under the Llama 3 Community License Agreement.

| Quantization Method | | Model Size | | | |
|---|---|---|---|---|---|
| Method | Bit | 1B | 3B | 8B | 70B |
| Base | 16-bit | ✓ | ✓ | ✓ | ✓ |
| AQLM | 2-bit | × | × | ✓ | × |
| AQLM-PV | 2-bit | ✓ | ✓ | ✓ | × |
| AWQ | 4-bit | ✓ | ✓ | ✓ | ✓ |
| BNB | 8-bit | ✓ | ✓ | ✓ | ✓ |
| | 4-bit | ✓ | ✓ | ✓ | ✓ |
| GPTQ | 8-bit | ✓ | ✓ | ✓ | × |
| | 4-bit | ✓ | ✓ | ✓ | × |
| | 3-bit | ✓ | ✓ | ✓ | × |
| | 2-bit | ✓ | ✓ | ✓ | × |
| HQQ | 8-bit | ✓ | ✓ | ✓ | ✓ |
| | 4-bit | ✓ | ✓ | ✓ | ✓ |
| | 3-bit | ✓ | ✓ | ✓ | ✓ |
| | 2-bit | ✓ | ✓ | ✓ | ✓ |
| QoQ | 4-bit | × | × | ✓ | × |
| Quanto | 8-bit | ✓ | ✓ | ✓ | ✓ |
| | 4-bit | ✓ | ✓ | ✓ | ✓ |
| | 2-bit | ✓ | ✓ | ✓ | ✓ |
| QuaRot | 8-bit | × | × | ✓ | × |
| | 4-bit | × | × | ✓ | × |

Table 2: Quantization techniques used in our reliability evaluation framework. ✓ corresponds to models included in the reliability assessment, and × corresponds to quantization-precision combinations we do not include.

| Task & Ability | Benchmark |
|---|---|
| **Language Modeling** | PTB (Marcus et al., 1993)
C4 (Raffel et al., 2020)
WikiText2 (Merity et al., 2016) |
| **Open-Ended QA Generation for General Knowledge & Dialog** | TriviaQA (Joshi et al., 2017)
CoQA (Reddy et al., 2019) |
| **In-Context Learning** | MMLU (Hendrycks et al., 2020)
CEval (Huang et al., 2023) |
| **Instruction Following** | HellaSwag (Zellers et al., 2019)
ARC (Clark et al., 2018) |
| **Reasoning** | PIQA (Bisk et al., 2020)
CommonsenseQA (Talmor et al., 2018) |
| **Understanding** | RACE (Lai et al., 2017) |

Table 3: Summary of the evaluated benchmarks. We provide the corresponding bit-level inference scalings in section 5 and section D.

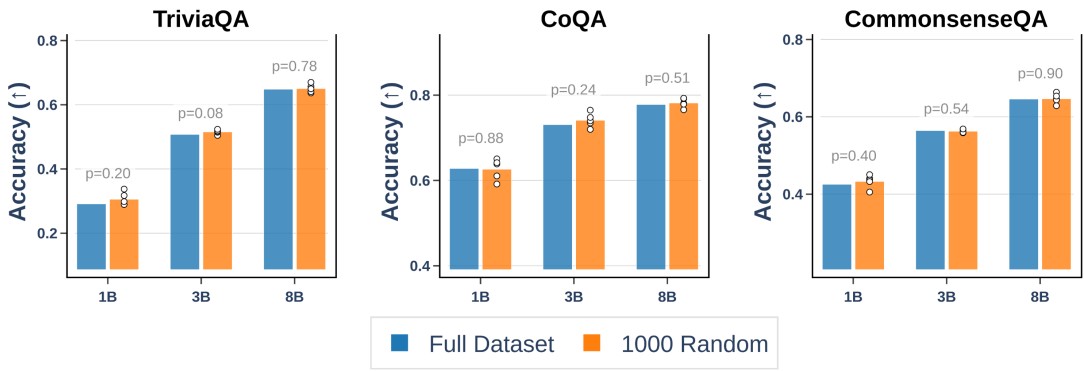

Figure 9: Comparison of the accuracy when evaluating on the full datasets versus evaluating on 1000 randomly sampled prompts (averaged over 5 random seeds).

## A.2 Evaluation datasets and generation

For the evaluation, we use two widely-used datasets for assessing the reliability of LLMs, TriviaQA (Joshi et al., 2017) and CoQA (Reddy et al., 2019). TriviaQA consists of 95,000 question-answer pairs to assess reading comprehension capabilities, while CoQA consists of 8,000 question-answer pairs. We further evaluate on the multiple-choice question answering CommonsenseQA (Lin & Och, 2004) dataset. We also evaluate on additional widely used benchmarks such as HellaSwag (Zellers et al., 2019), MMLU (Hendrycks et al., 2020), and Arc-Easy (Clark et al., 2018). We summarize all evaluation benchmarks in table 3. For all datasets, we limit the evaluation to 1000 uniformly sampled prompts, due to the increased number of evaluation combinations as we consider four base models, eight quantization models, and five different bit precision levels. We randomly sample 1000 samples from the datasets for 5 random seeds, and evaluate the accuracy. We compare the accuracies over the sampled sets with the accuracy averaged across the entire datasets. We show in fig. 9 that the accuracy on the subsets is close to the full dataset evaluation. During generation, we limit the generated sequence length to 20 tokens per input prompt. During the generation, we use multinomial sampling with a temperature parameter of 0.7.

We conduct our experiments using GTX 1080 Ti (11GB) and A100 (40GB) GPUs. For smaller models, specifically all 1B and 3B models (excluding AQLM and GPTQ variants), we use a single GTX 1080 Ti

| Leetspeak Character Substitutions | | | |
|---|---|---|---|
| A → @ | a → @ | B → 8 | b → 6 |
| C → ( | c → © | D → ) | d → δ |
| E → 3 | e → € | G → 6 | g → 9 |
| H → # | I → 1 | i → ! | J → 7 |
| K → \|< | L → £ | l → \| | O → 0 |
| o → ° | P → ? | R → ® | S → 5 |
| s → § | T → 7 | w → ω | X → × |

Table 4: Examples of leetspeak character perturbations.

GPU. Medium-sized models, including the 1B and 3B AQLM variants, GPTQ, the 8-bit versions, and all 8B models, are evaluated on a single A100 GPU. For the largest 70B models, we use five A100 GPUs in parallel to handle loading and evaluation efficiently. For the evaluation runtime given a single perturbation (or no perturbation), the average evaluation time is approximately 5 minutes on the GTX 1080 Ti for the small models, and 4 minutes on the A100, and roughly 24 minutes for the 70B models on the five A100 GPUs.

## B    Additional details on the perturbations

In addition to the character-level and word-level perturbations described in section 3.2, we provide additional details on additional perturbations in the following:

- Deletion: For character-level perturbation, we randomly remove characters from the sentence; at the word level, we remove words from text, primarily targeting filler words (e.g., "and", "to", "also", "actually") to preserve the important part of the prompt (Moradi & Samwald, 2021). We use 508 filler words in total.

- Leetspeak: Substitutes characters with visually similar numerals or symbols (like replacing 'e' with '3'). We provide additional examples in table 4. We use 93 Leetspeak examples.

- Noise: Inserts punctuation or digits as noise characters distributed randomly throughout the text, using a configurable noise character set.

- Replacement: Substitutes characters with adjacent keyboard alternatives to simulate realistic typos, using a keyboard layout mapping maintaining case sensitivity (Ackerman et al., 2024).

- Swapping: Exchanges adjacent characters to simulate typographical errors; at the word level, exchanges adjacent words while preserving punctuation and maintaining the question-answer prompt structure (Moradi & Samwald, 2021; Wang et al., 2024).

- Repetition: Duplicates characters at random positions; at the word level, duplicates words while handling punctuation preservation and maintaining overall text structure (Moradi & Samwald, 2021).

- Case Change: Alters letter case patterns, toggling between uppercase and lowercase only for alphabetic characters (Moradi & Samwald, 2021).

- Emoji: A novel perturbation type that inserts emoji characters within text, distributing them across words using a configurable emoji set. We use 115 emoji characters in total.

- Slang: Inserts internet slang expressions (like "lol", "rofl", "imo") using a predefined dictionary of common terms, distributing insertions randomly throughout the text.

- Translation: Translates words and phrases through Google Translate API across seven languages (Spanish, French, German, Italian, Russian, Chinese, Japanese). As opposed to the semantic-level translation perturbation introduced in Zhu et al. (2024), our method does not translate phrases back to English, simulating multi-lingual user outputs.

- Insertion: Randomly inserts characters between existing characters at the character level; at the word level, adds contextually relevant words using RoBERTa masked language model to predict insertions while preserving text coherence. This approach is methodologically similar to the synonym insertion implemented by Moradi & Samwald (2021), but only adds words and does not change them.

For the leetspeak character perturbations, we provide examples in table 4. Each perturbation is applied at varying intensity levels of 4 and 16 to systematically evaluate model robustness across multiple intensities. We note that for the word perturbations presented in fig. 2, the number of perturbed words in the input prompt corresponds to the minimum between the perturbation intensity and the length of the input prompt.

## C  Additional Efficiency Metrics

In this section, we examine how the total model bits metric relates to several commonly used efficiency measures, including inference memory, latency, and throughput. A quantitative comparison of these metrics is provided in table 1 for different models from the Qwen3 model family. As introduced in section 4, the total number of bits $B$ is defined as the product of the number of model parameters and their weight precision (in bits). This quantity corresponds directly to the model's storage memory. The total model bits further maps to the inference memory, measured as:

$$\text{Inference memory} = P \times \text{Avg-Bitwidth} + \text{Overhead}, \tag{8}$$

where $P$ is the total number of parameters, and the Overhead term accounts for the additional memory used during inference (e.g., KV cache, activation). In practice, the overhead is generally negligible relative to the storage memory, which implies that inference memory scales almost linearly with the total model bits.

For inference latency, the relationship with total model bits is less direct but still significant. The overall latency is determined by two phases: (1) the pre-fill phase and (2) the decode phase. During prefill, the model processes the input sequence and constructs the key–value (KV) cache. This stage is typically compute-intensive and can fully utilize GPU compute units. During the decode phase, the model generates output tokens sequentially. Each token is predicted based on the previously generated tokens and the information stored in the KV cache from the prefill stage. Lower-precision numbers can improve speed because they reduce the amount of data that needs to be moved in memory and allow computations to be executed more quickly. For instance, on modern GPUs such as H100, 8-bit operations can achieve significantly higher FLOPS than 32-bit operations. However, this improvement also depends on specialized hardware support [1]. For future work, it would be interesting to investigate scaling laws as a function of latency, as it depends not only on the model, but also on the hardware. We note that we use a batch size of 1 in our setup (see section A.1) and compute the corresponding token latency and throughput.

The throughput defines the number of output tokens that can be predicted per time unit. For sequential, non-parallel decoding, such as autoregressive LLM, latency and throughput are directly coupled. More specifically, for a batch size of 1, the throughput corresponds to $\frac{1}{\text{Latency}}$. Since decoding is inherently sequential and the setup is compute-bound, improving the latency directly improves throughput.

In table 1, we compare the different efficiency metrics for models from the Qwen3 model family. Specifically, we select models with 4B, 8B, and 16B parameters. For the quantization, we use BitsandBytes (Dettmers et al., 2022) and quantize the full-precision models to 4 and 8 bits.

---

[1] https://resources.nvidia.com/en-us-gpu-resources/h100-datasheet-24306

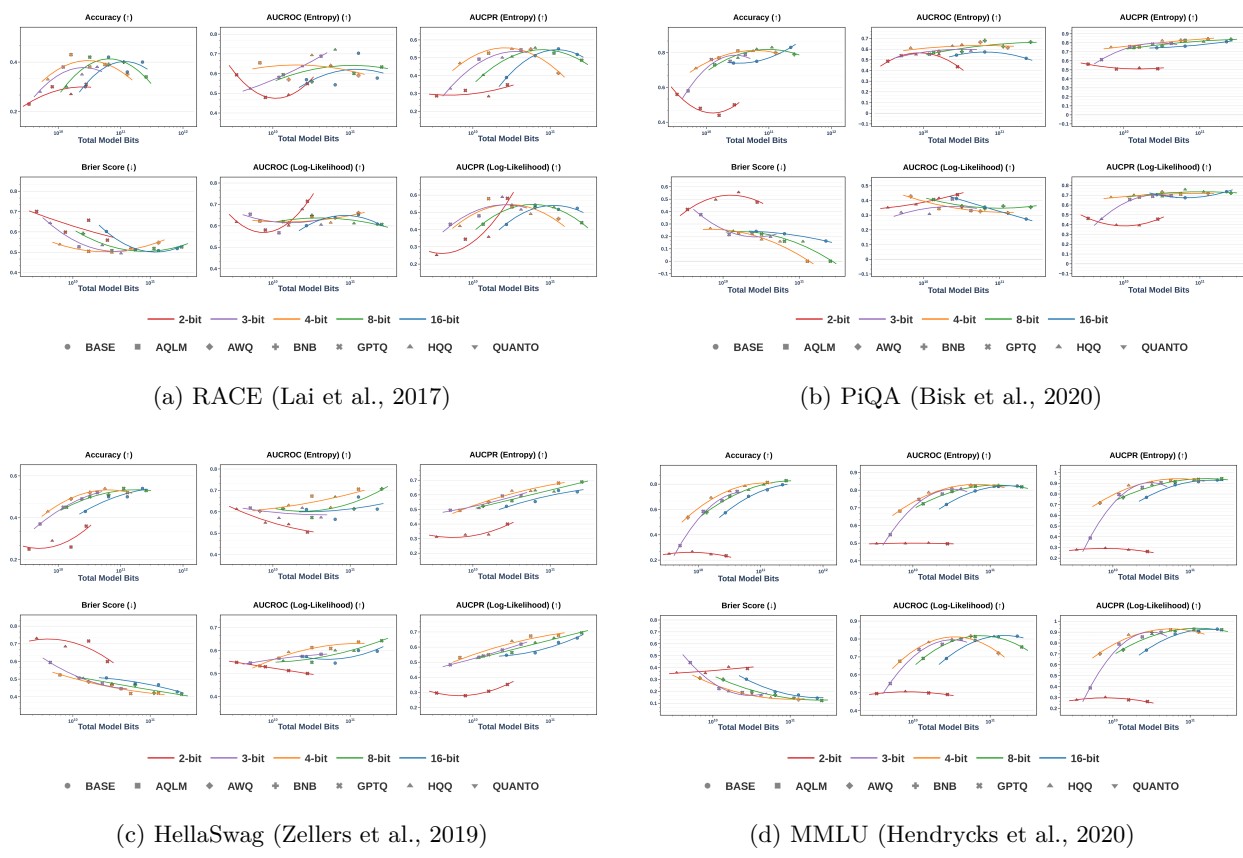

(a) RACE (Lai et al., 2017)

(b) PiQA (Bisk et al., 2020)

(c) HellaSwag (Zellers et al., 2019)

(d) MMLU (Hendrycks et al., 2020)

Figure 10: Bit-level scaling trends of Qwen3 models across four benchmarks.

## D Reliability Scaling Trends Across Different Benchmarks

**Reliability scaling trends of Qwen3 models** We evaluate four base models from the Qwen3 model family, with sizes of 4B, 8B, 14B, and 32B, and their corresponding quantized models using 6 different PTQ methods. We provide results in fig. 10 We evaluate the reasoning abilities on the PiQA benchmark in fig. 10b, and the understanding abilities on RACE in fig. 10a. For instruction-following assessment, we use the HellaSwag benchmark and provide results in fig. 11b. For in-context learning, we evaluate on the widely-used MMLU dataset in fig. 10d. Across different datasets, model sizes, and bitwidths, we observe a sweet spot at 4-bit quantization. Both GPTQ and HQQ provide the best performance for 4-bit quantization. We further note that for Qwen3 models, 3-bit quantization can provide favorable performance in terms of zero-shot performance and reliability.

**Reliability scaling trends of LLaMA-3 models** In addition to the performance and reliability bit-level inference scalings provided in section 5, we extend the evaluation to in-context learning and instruction-following tasks to assess the emergent abilities of quantized LLMs from the LLaMA-3 model family. For in-context learning, we provide the bit-level inference scaling plots under fig. 11a for the MMLU task. For instruction-following, we evaluate on HellaSwag in fig. 11b and ARC in fig. 11c. We further assess the open-ended generation on CoQA in fig. 11d, which is a conversational question-answering task that tests the dialog understanding capabilities of models. Additional bit-level scalings on the instruction-tuned LLaMA models are presented in fig. 12. For the 2-bit precision, AQLM-PV provides the best zero-shot accuracy. For 3-bit models, HQQ outperforms GPTQ across different metrics. For 4-bit models, both AWQ (Lin et al., 2024a) and GPTQ (Frantar et al., 2022) generally provide the best downstream task performance and reliability.

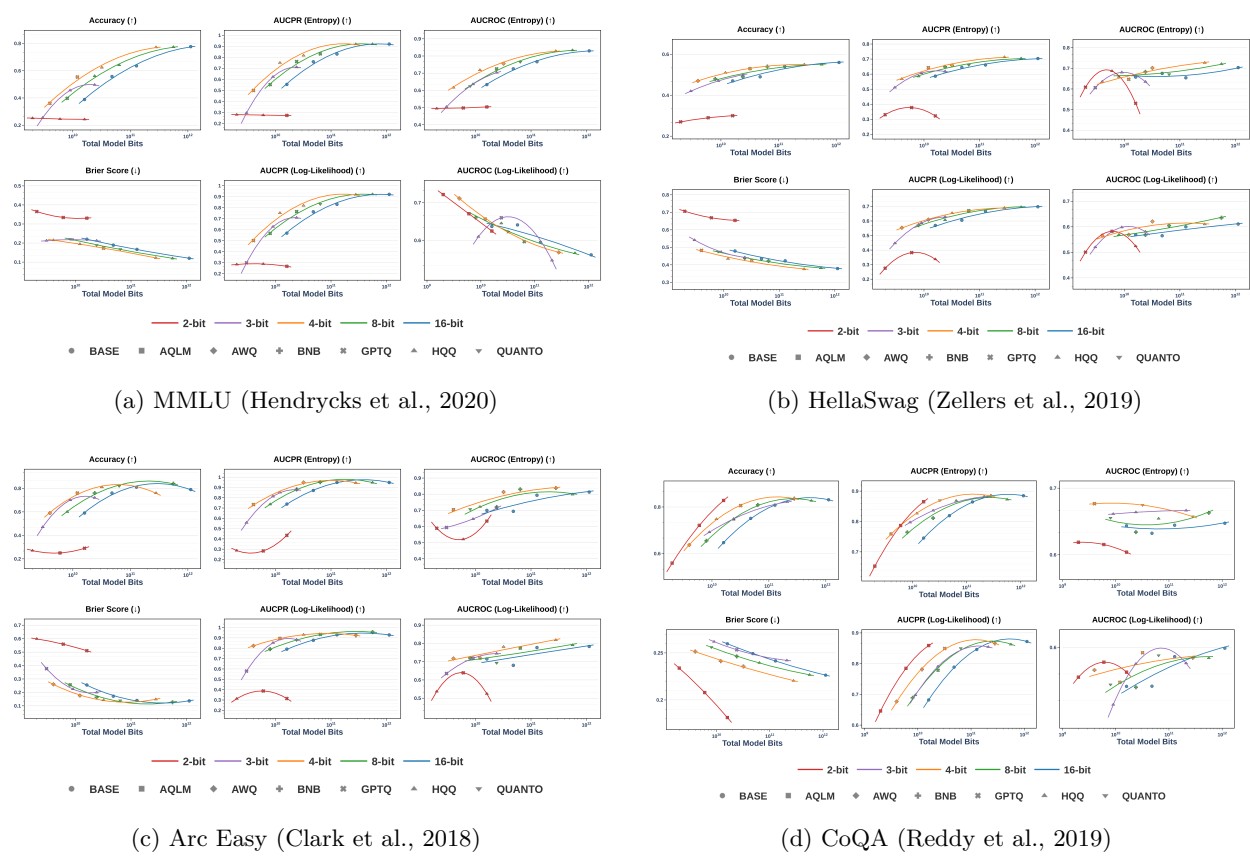

Figure 11: Bit-level scaling trends of LLaMA models across four benchmarks.

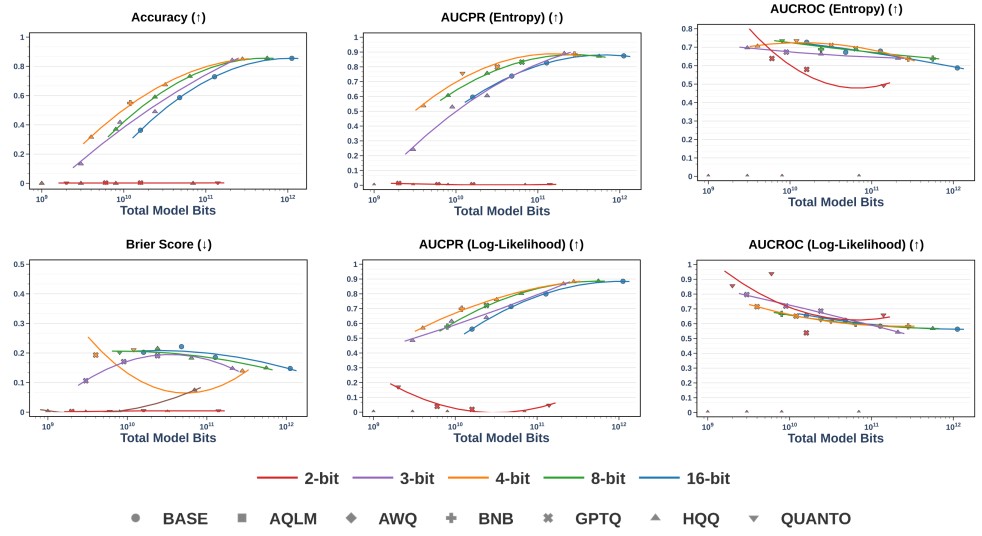

Figure 12: Scaling behavior of the reliability metrics of the **instruction-tuned** LLaMA models on **TriviaQA**.

| Model | Intensity | Metrics | | |
|---|---|---|---|---|
| | | Accuracy | AUCROC (Entropy) | AUCPR (Entropy) |
| LLaMA-3.2-1B | 0 | $0.338 \pm 0.000$ | $0.683 \pm 0.000$ | $0.518 \pm 0.000$ |
| | 4 | $0.192 \pm 0.050$ | $0.636 \pm 0.029$ | $0.264 \pm 0.084$ |
| | 16 | $0.078 \pm 0.077$ | $0.578 \pm 0.069$ | $0.098 \pm 0.118$ |
| LLaMA-3.2-3B | 0 | $0.549 \pm 0.000$ | $0.722 \pm 0.000$ | $0.746 \pm 0.000$ |
| | 4 | $0.384 \pm 0.064$ | $0.689 \pm 0.020$ | $0.534 \pm 0.083$ |
| | 16 | $0.189 \pm 0.126$ | $0.608 \pm 0.076$ | $0.234 \pm 0.170$ |
| LLaMA-3-8B | 0 | $0.691 \pm 0.000$ | $0.715 \pm 0.000$ | $0.847 \pm 0.000$ |
| | 4 | $0.519 \pm 0.069$ | $0.691 \pm 0.022$ | $0.687 \pm 0.065$ |
| | 16 | $0.283 \pm 0.156$ | $0.640 \pm 0.051$ | $0.357 \pm 0.189$ |
| LLaMA-3-70B | 0 | $0.728 \pm 0.000$ | $0.503 \pm 0.000$ | $0.729 \pm 0.000$ |
| | 4 | $0.706 \pm 0.037$ | $0.557 \pm 0.036$ | $0.736 \pm 0.036$ |
| | 16 | $0.503 \pm 0.145$ | $0.595 \pm 0.063$ | $0.544 \pm 0.136$ |

Table 5: Performance across different perturbation intensities on the TriviaQA dataset

| Base Model | Bit Width | Accuracy | AUCROC (Entropy) | AUCPR (Entropy) | AUCPR (Log-Lik.) |
|---|---|---|---|---|---|
| 1B | 2 | AQLM | AQLM | AQLM | AQLM |
| 1B | 4 | AWQ | GPTQ | AWQ | GPTQ |
| 1B | 8 | GPTQ | Quanto | GPTQ | BNB |
| 3B | 2 | AQLM | AQLM | AQLM | AQLM |
| 3B | 3 | HQQ | HQQ | HQQ | HQQ |
| 3B | 4 | HQQ | GPTQ | HQQ | GPTQ |
| 3B | 8 | Quanto | BNB | Quanto | Quanto |
| 8B | 2 | AQLM | AQLM | AQLM | AQLM |
| 8B | 3 | HQQ | HQQ | HQQ | HQQ |
| 8B | 4 | HQQ | GPTQ | AWQ | HQQ |
| 8B | 8 | HQQ | Quanto | HQQ | GPTQ |
| 70B | 3 | HQQ | HQQ | HQQ | HQQ |
| 70B | 4 | AWQ | AWQ | AWQ | AWQ |
| 70B | 8 | HQQ | BNB | Quanto | HQQ |

Table 6: Recommendation list of the best quantization methods per model size and bit width across different evaluation metrics on the unperturbed TriviaQA dataset.

# E Reliability Scaling Trends on perturbed Benchmarks

In this section, we provide additional results to show the scaling behavior of the accuracy, the quality of the uncertainty estimates, the log-likelihood, and the calibration using the perturbed prompts. In particular, we provide the results for the Word Slang perturbation in fig. 13 and the Leetspeak perturbation in fig. 14. For all experiments, we fit a log-quadratic function per bit width, and only show the best-performing model for every bit level. We further report the mean and standard deviation of the accuracy and the reliability metrics of the four base LLaMA-3 models, under unperturbed and perturbed input prompts, in table 5. Additionally, in table 7, we provide a list of the best quantization methods per base model and precision for two different perturbation intensities. For the 2-bit precision, AQLM-PV provides the best performance on the perturbed datasets. For 3-bit models, HQQ outperforms GPTQ across different metrics. For 4-bit models, HQQ (Badri & Shaji, 2023) and GPTQ (Frantar et al., 2022) generally provide the best downstream performance and reliability under perturbation, followed by AWQ (Lin et al., 2024a).

| Base Model | Bit Width | Accuracy | AUCROC (Entropy) | AUCPR (Entropy) | AUCPR (Log-Lik.) |
|---|---|---|---|---|---|
| 1B | 2 | AQLM | AQLM | AQLM | - |
| 1B | 4 | AWQ | GPTQ | AWQ | HQQ |
| 1B | 8 | HQQ | GPTQ | HQQ | BNB |
| 3B | 2 | AQLM | AQLM | AQLM | - |
| 3B | 3 | HQQ | HQQ | HQQ | HQQ |
| 3B | 4 | HQQ | GPTQ | HQQ | Quanto |
| 3B | 8 | Quanto | BNB | Quanto | HQQ |
| 8B | 2 | AQLM | AQLM | AQLM | - |
| 8B | 3 | HQQ | HQQ | HQQ | HQQ |
| 8B | 4 | AWQ | GPTQ | AWQ | HQQ |
| 8B | 8 | HQQ | Quanto | GPTQ | Quanto |
| 70B | 3 | HQQ | HQQ | HQQ | HQQ |
| 70B | 4 | AWQ | AWQ | AWQ | HQQ |
| 70B | 8 | HQQ | BNB | HQQ | HQQ |

Table 7: Recommendation list of the best quantization methods per model size and bit width across different evaluation metrics on the **perturbed** TriviaQA dataset. We average the performance over all 15 perturbations with an intensity equal to **4**.

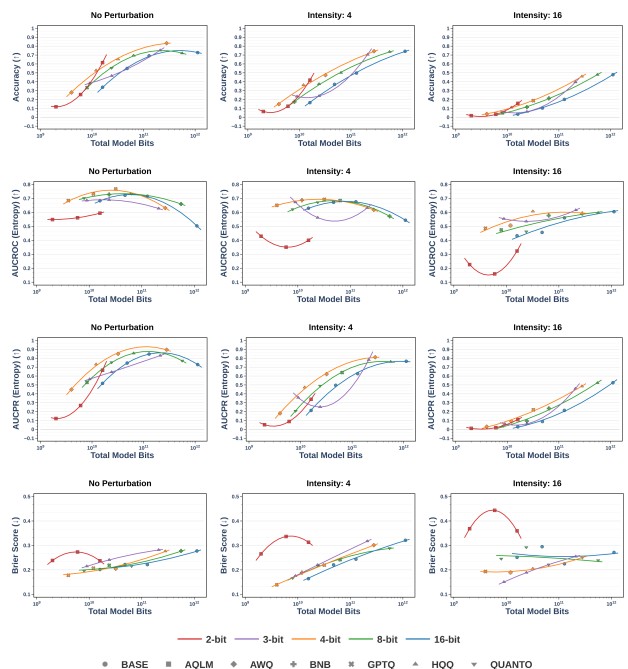

Figure 13: Scaling behavior of the accuracy and various uncertainty quantification and calibration metrics on the unperturbed TriviaQA dataset, as well as the perturbed prompts using the **word slang** perturbation using two perturbation intensities of 4 and 16.

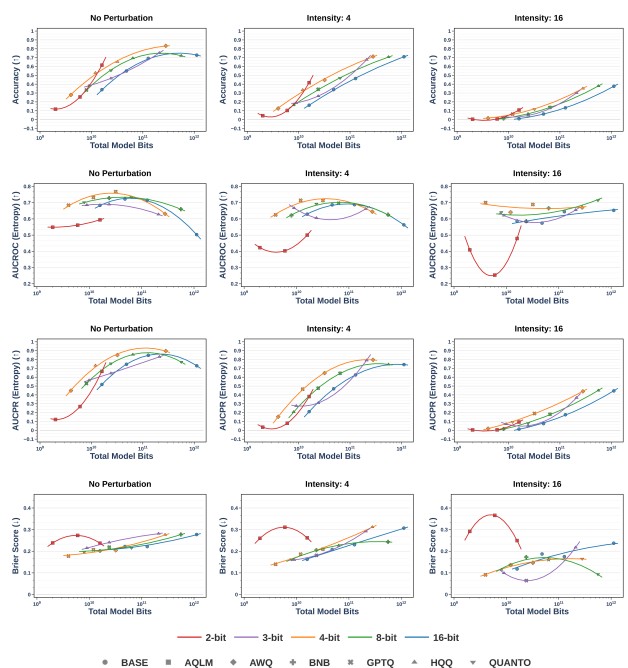

Figure 14: Scaling behavior of the accuracy and various uncertainty quantification and calibration metrics on the unperturbed TriviaQA dataset, as well as the perturbed prompts using the **Leetspeak** perturbation using two perturbation intensities of 4 and 16.

## F   Additional Ablations

In this section, we provide additional ablations across temperature used for sampling and the length of the generated sequences. We note that we limit the evaluation to 100 samples from the TriviaQA dataset. We first examine the bit-level inference scalings under different decoding strategies. Specifically, evaluate the open-ended generation of base and quantized LLMs on TriviaQA, where we sample with various temperature values in 0.2, 0.7, 1.0. We present the qualitative results in fig. 15. Overall, the performance and reliability metrics exhibit consistent trends across temperature settings. In particular, accuracy increases with the total number of model bits, with 4-bit models achieving the strongest performance. For reliability, we observe a pronounced peak for 4-bit quantized models. However, as the temperature increases to 0.7 and 1.0, 3-bit quantized models yield the best Brier scores, given a fixed total model bits. In fig. 3, we truncate the generation of models to 20 tokens. We explore longer generations using an increased number of tokens. We present results in fig. 16.

## G   How does Quantization-Aware Training affect the scaling trends?

While our study primarily focuses on state-of-the-art post-training quantization techniques for LLMs, we investigate in the following how quantization-aware training (QAT) influences the performance and reliability scaling trends. QAT is especially compelling for extreme quantization regimes, as post-training quantizations often result in substantial performance degradation, as shown in **??**.

In the following, we examine the recent EfficientQAT (Chen et al., 2025) method, which consists of two stages: block-wise training of all model parameters followed by end-to-end training of the quantization parameters. We apply QAT to three LLaMA-3 models with 7B, 13B, and 70B parameters and evaluate 4-bit and 2-bit quantized models as well as their full-precision counterparts. We present qualitative results of the bit-level inference scalings in fig. 17. We find that 4-bit quantized models generally offer favorable zero-shot performance and reliability for a fixed total bit budget. EfficientQAT (Chen et al., 2025) achieves impressive

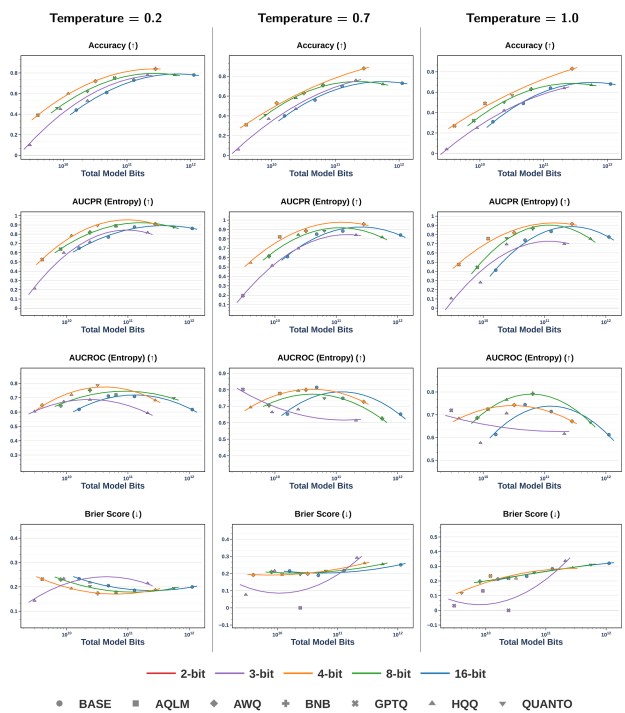

Figure 15: Bit-level inference scaling trends on TriviaQA for various temperature values for sampling.

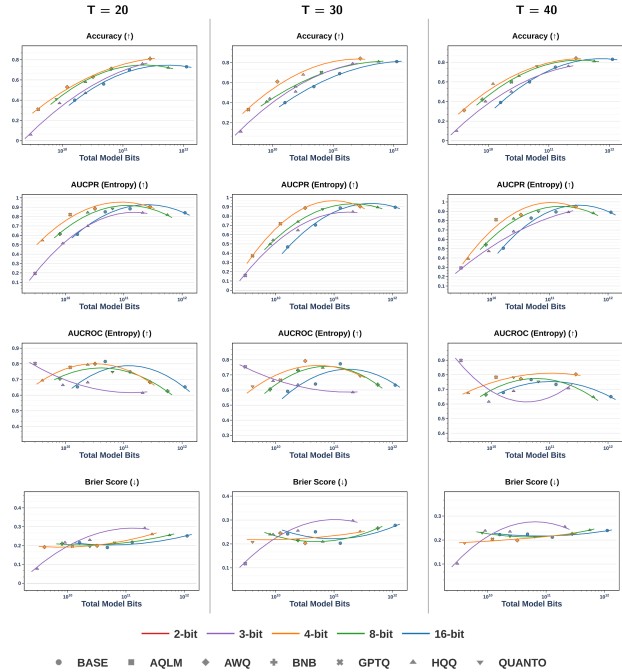

Figure 16: Bit-level inference scaling trends on TriviaQA for different numbers of output tokens.

results in 2-bit scenarios, improving the zero-shot performance and reliability scaling trends compared to PTQ approaches. However, a performance gap remains compared to higher bitwidths.

We limit the evaluation to the EfficientQAT approach due to time and resource constraints. Extending the bit-level reliability scaling study to different QAT approaches (Liu et al., 2024; Ma et al., 2024; Xu et al., 2024a) and performing a more thorough comparison across different model backbones, tasks, and bit precisions could yield novel insights, which we leave for future work. While QAT techniques are promising for extremely low bit-widths, there remains a need for more efficient and practical approaches. For example, while BitNet b1.58 (Ma et al., 2024) achieves nearly lossless ternary quantization, it requires retraining LLMs from scratch on the entire pre-trained dataset. This makes it impractical for huge models and restricts its validation to 3B models trained on 100B tokens, limiting its applicability for comprehensive scaling law studies across varying model sizes.

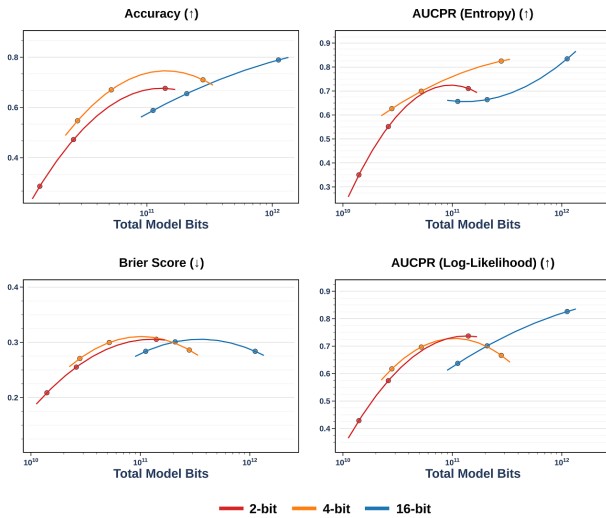

Figure 17: Bit-level scaling trends of quantized LLaMA models using EfficientQAT (Chen et al., 2025).

## H   Are the bit-level inference scalings consistent across different quantization methods?

To assess whether the observed bit-level inference scaling is specific to a single quantization method, we compare several 4-bit PTQ techniques across different model families. In particular, we evaluate HQQ, GPTQ, BNB, AWQ, and Quanto, and compare them against the corresponding 16-bit full-precision baselines. The results show that the overall trends are largely consistent across quantization methods: performance improves with the total number of model bits, while reliability-related metrics exhibit non-linear trends. Importantly, the 4-bit models often match or outperform the full-precision baselines in calibration and uncertainty-based reliability metrics, suggesting that the reliability peak is not tied to a specific quantization method.

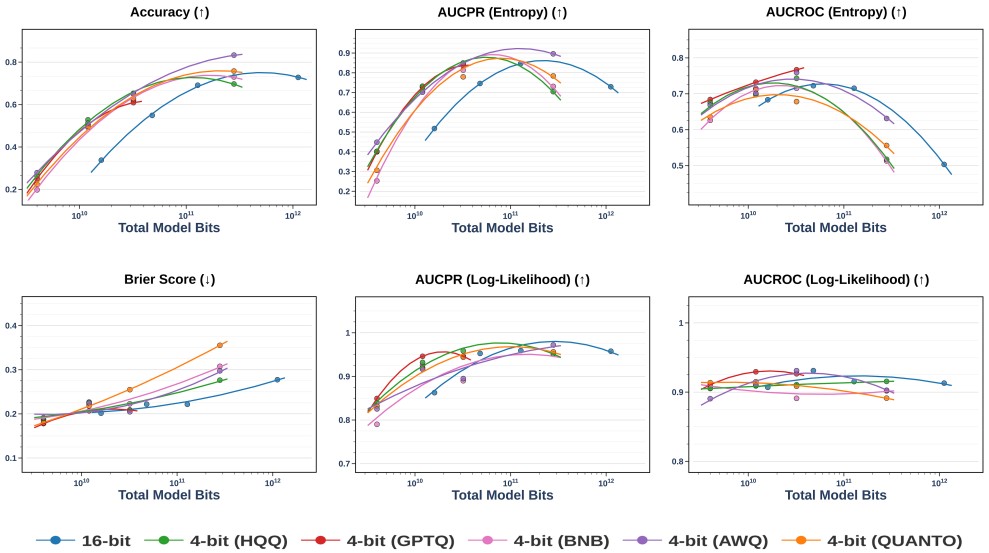

Figure 18: Bit-level inference scaling trends of LLaMA models under different 4-bit quantization methods.

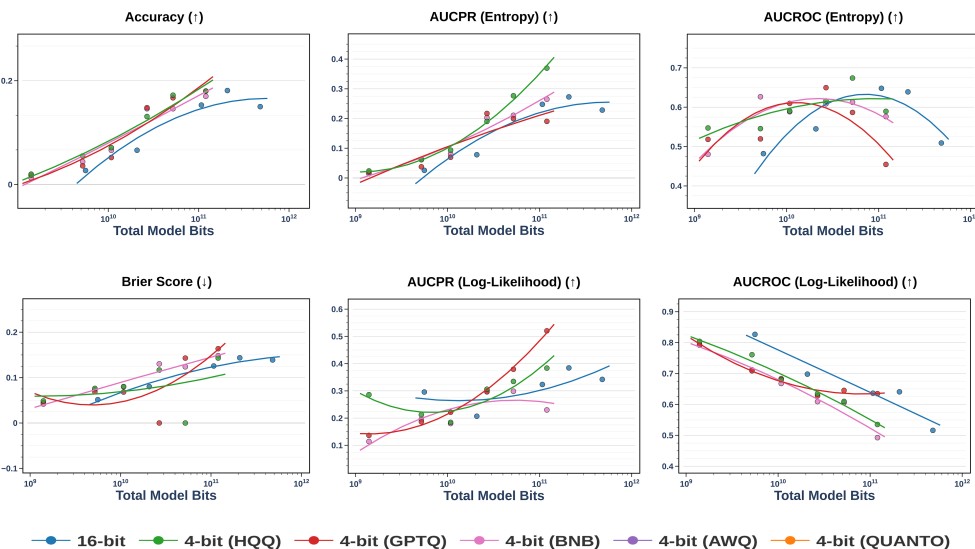

Figure 19: Bit-level inference scaling trends of OPT models under different 4-bit quantization methods.

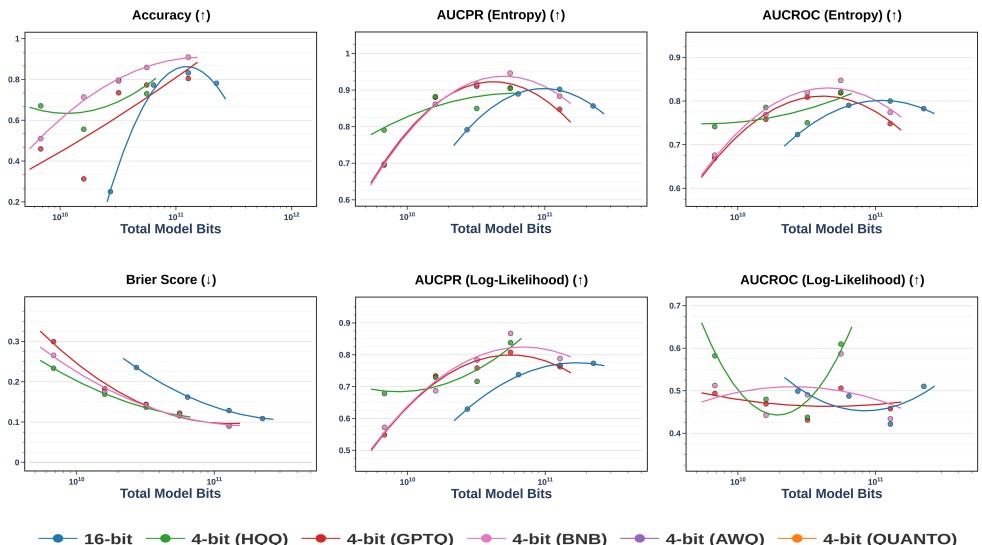

Figure 20: Bit-level inference scaling trends of Qwen3 models under different 4-bit quantization methods.

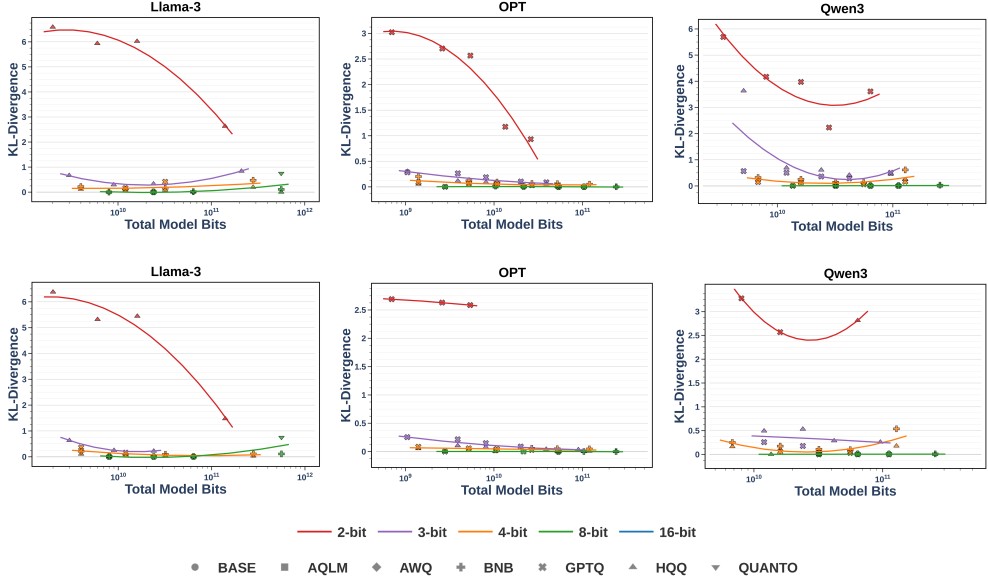

Figure 21: Additional KL-divergence analysis across LLaMA-3, OPT, and Qwen3 models on Wikitext (top) and C4 (bottom) datasets.

# I    Additional KLD analysis

To further analyze the behavioral shift induced by quantization, we extend the KL-divergence analysis beyond the LLaMA family. In particular, we compare the token-level predictive distributions of quantized models against their corresponding full-precision base models across three model families: LLaMA-3, OPT, and Qwen3. Please refer to fig. 21. We evaluate multiple quantization precisions, including 2-, 3-, 4-, and 8-bit models, on two different benchmarks. This allows us to assess whether the observed relationship between quantization precision and behavioral shift is consistent across architectures and tasks.

