# OpenReview forum: "Reliability Scaling Laws for Quantized Large Language Models"
_TMLR — Accepted by TMLR_

### Review · Reviewer_XWqU · 2026-05-01

**Summary Of Contributions:**

The authors investigate model reliability as a function of bits (using bits to enable abstracting away model size vs quantization level). The authors claim that unlike performance, reliability does not scale monotonically with number of bits.

**Audience:**

Yes

**Audience Explanation:**

The authors provide an interesting empirical finding on the trend of reliability metrics with model scale (in bits), which has been under-explored.

**Claims And Evidence:**

No

**Claims Explanation:**

While Figures 1,7,8 show a clear non-monotonic trend, I have a question for clarification:

I am interpreting the figure as follows: on a single curve, each data point is a different model, that has been quantized down to the bit-level corresponding to that line. However, for a given line/quantization level, I see different quantization methods? E.g., for 8-bit quantization (green line in Figure 1), I see both Qunato and BNB are used. Whereas for 4-bit I see GPTQ and AQLM are used. Why this inconsistency? To have more confidence in the claim of non-monotonicity and I would have expected less mixing between these. E.g., I would expect see the whole plot performed with a single quantization method, then perhaps repeated for a number of other methods in the appendix.

**Requested Changes:**

- [critical] Fixing Figures 1,7,8 as I discussed above.
- [critical] The authors tend to mix two claims throughout the paper "studying the reliability of quantized models, which has been under-explored" and presenting a "scaling law for reliability metrics". For example, the abstract focuses on the former claim whereas section 3.3 presents the latter claim. The separation of these claims is not very clear and the work could benefit from treating them a bit separately, or at the very least making them more explicit.
- [minor] The last paragraph of page 9 only answers why 4-bit is better than 2,3-bit. But we see 8-bit is also stable across this range. So this is a bit misleading as a title perhaps. I was expecting an explanation for why the 4-bit curve outperforms the 8-bit curve given the paragraph title.

---

> ### Author Response · Authors · 2026-05-22
>
> We thank the reviewer for their feedback. We have revised our manuscript to address the requested changes.
>
> > Clarification on Figures
>
> We thank the reviewer for raising this point. For each bit width and model size, we selected the best-performing quantization method and plotted that single point. This choice was motivated by two considerations: clarity and readability of the figures, given the large number of model/quantization method/bitwidth combinations, and practical relevance; in a real inference setting, a user selecting among quantization methods at a fixed bit width would choose the best-performing variant.
>
> However, we agree that this design choice makes it difficult to assess whether the observed non-monotonicity is stable across individual settings. As suggested by the reviewer, we have updated our scaling figures to include all quantized models at all bit widths, with a separate fitted log-quadratic curve for each bit width. The non-monotonic reliability trend and the 4-bit peak remain consistent. Please refer to the updated manuscript.
>
> > The authors tend to mix two claims throughout the paper
>
> We agree that the original manuscript did not sufficiently distinguish between two related but distinct contributions: the reliability evaluation of quantized LLMs and the empirical study of reliability scaling trends. In the revised manuscript, we make this distinction explicit in both the abstract and introduction. Specifically, we present the reliability evaluation as the framework that defines the dimensions we measure, including uncertainty, calibration, and robustness to natural perturbations. In contrast, the scaling analysis studies how these reliability metrics vary as a function of total model bits. We also restructure the manuscript accordingly: Section 3 now introduces the reliability assessment framework, while Section 4 introduces the empirical bit-level scaling laws.
>
> > The last paragraph of page 9 only answers why 4-bit is better than 2,3-bit
>
> We explain the reliability peak at 4-bit in the following: First, the KL-divergence analysis (Figure 6) explains why 4-bit precision outperforms 2 and 3-bit precisions, since the behavioral shift of 4-bit models to their full-precision counterparts is significantly smaller than that of 2- and 3-bit models. For the 4-bit vs 8-bit comparison, the key additional factor is model capacity under a fixed total-bit budget. A 4-bit model matched to an 8-bit model in total bits has roughly twice the number of parameters. Our results suggest that, in this regime, the benefit of increased model capacity can outweigh the benefit of higher weight precision: 4-bit models remain sufficiently close to their full-precision counterparts while benefiting from a larger parameter count. Below 4 bits, this trade-off changes: the behavioral shift induced by aggressive quantization becomes too large to be compensated by additional parameters. Please refer to our updated manuscript, where we clarify this point.

---

### Review · Reviewer_G7qU · 2026-05-01

**Summary Of Contributions:**

This paper explores the performance of quantized Large Language Models (LLMs) on perturbed inputs. More specifically, by focusing on six state-of-the-art (sota) quantization techniques, it aims at identifying whether there exists a precision maximizing the reliability of the LLMs, providing an optimal trade-off with computational efficiency. This reliability is measured thanks to relevant uncertainty metrics (e.g. entropy), and robustness to natural perturbations. The authors claim the following contributions:
1) Thanks to experiments on different models with six sota quantization techniques and 15 natural perturbations applied to the inputs, they find that 4-bit precision offers the best trade-off between reliability and efficiency.
2) They further investigate why the 4-bit precision provides this optimal trade-off by studying the Kullback-Leibler Divergence, and observe that the last significant behavorial shift when increasing the precision appears at 4 bits.

**Audience:**

Yes

**Audience Explanation:**

As LLMs, and in particular their efficiency and reliability, are a very hot topic, TMLR's audience would indeed be interested in knowing the findings of this paper. Moreover, the authors discuss the related works, as well as the limitations of their work.

**Claims And Evidence:**

Yes

**Claims Explanation:**

The claims are supported by an empirical analysis on six sota quantization techniques, six datasets from various tasks and four different LLMs. The authors also explore 15 natural perturbations for the robustness analysis. I am not familiar with this literature, but it does seem to me that the provided empirical evidence is rather convincing.

**Requested Changes:**

N/A

---

> ### Author Response · Authors · 2026-05-22
>
> We thank the reviewer for their positive assessment of our work and for recognizing the relevance of studying reliability and robustness in quantized LLMs. We appreciate the reviewer’s acknowledgment that our empirical evaluation across multiple quantization techniques, datasets, models, and natural perturbations provides convincing support for our claims. Please refer to our updated manuscript, which incorporates revisions addressing the comments raised by Reviewers XWqU and w7xt.

---

### Review · Reviewer_w7xt · 2026-05-11

**Summary Of Contributions:**

The paper empirically studies how quantization affects LLM reliability beyond standard accuracy and perplexity. It evaluates multiple post-training quantization methods across different bit settings, measuring uncertainty, calibration-related metrics, and robustness to character- and word-level input perturbations. The main finding is that the reliability scaling appears nonlinear, with 4-bit quantized models often showing the best reliability–efficiency trade-off.  This paper targets an interesting topic and presents a relatively broad empirical scope across several quantization methods and bit-widths. However, the claim about scaling is stronger than what the evidence supports.

**Audience:**

Yes

**Audience Explanation:**

While prior work on quantization scaling laws has mainly focused on accuracy or perplexity under different bit budgets, this paper studies reliability. This perspective is quite novel and interesting. Prior work has studied LLM quantization extensively, but the reliability and scaling law perspectives have not been as comprehensively explored. The main finding about the nonlinear scaling is interesting and has clear potential value for both the quantization and reliability/trustworthy communities.

**Claims And Evidence:**

No

**Claims Explanation:**

- Limited generalization weakens the broad claim on “reliability scaling law”: The core experiments remain centered on LLaMA-family models and QA-style evaluations, while OPT and Qwen3 results are presented more as additional qualitative evidence than as a fully symmetric cross-architecture study. The paper also uses an incomplete method–bitwidth coverage and reports only the best-performing model per bitwidth in several scaling plots, which makes it difficult to assess whether the observed 4-bit peak is stable across quantization methods. Thus, the current evidence supports a useful empirical trend within the studied settings, but not yet a broadly valid reliability scaling law.
- 4-bit reliability peak lacks a sufficient explanation: This work is largely empirical, with a limited mechanistic explanation for why 4-bit quantization should be optimal and how it would advise future study. The KL-divergence analysis helps describe behavioral shift, but it does not fully explain why 4-bit models should outperform 8-bit or 16-bit models on reliability. Also, the KL analysis is limited to three LLaMA models and does not cover the full range of architectures, tasks, perturbations, or quantization methods used to support the broader claim. Moreover, the scaling curves are fit on and report only the best-performing model per bit-width; the observed peak may reflect method/model selection effects rather than a principled reliability mechanism.

**Requested Changes:**

- The authors should either substantially qualify the scope of the claim or provide stronger cross-setting evidence.
- The authors should provide more statistical evidence to support that the 4-bit peak is statistically stable across datasets, perturbation types, random seeds, decoding settings, and quantization methods. Also, the current mechanism study remains too weak and empirical. The authors should go beyond post-hoc behavioral-shift/KL analysis

---

> ### Author Response · Authors · 2026-05-22
>
> We are pleased with the positive feedback on our work, particularly noting its motivation and highlighting the novelty of the reliability study, and its relevance to the fields of quantization and reliability.
>
> >The paper uses an incomplete method–bitwidth coverage and reports only the best-performing model per bitwidth in several scaling plots
>
> We thank the reviewer for raising this point. Our goal was to keep the scaling plots readable and to focus on practically relevant quantized models. However, we agree that this presentation can make it difficult to assess whether the observed trends are stable under different settings. In our updated manuscript, we have updated the scaling figures to include all available quantized models at each precision, rather than only the best-performing model. We fit a scaling curve for each precision using all corresponding quantized models. Importantly, our main findings remain consistent: reliability still exhibits a non-monotonic trend with total model bits, and the 4-bit regime remains favorable compared to both lower-bit and higher-bit settings. Please refer to Figures 1, 4, 5, and 6.
>
> In addition, following the reviewer's suggestion, we check whether the observed 4-bit peak is stable across quantization methods by comparing several 4-bit PTQ techniques across different model families (LLaMA, OPT, Qwen3). Please refer to Appendix H in our updated PDF. In particular, we evaluate HQQ, GPTQ, BNB, AWQ, and Quanto, and compare them against the corresponding 16-bit full-precision baselines. The results in Figures 18, 19 and 20 show that the overall trends are largely consistent across quantization methods: performance improves with the total number of model bits, while reliability-related metrics exhibit non-linear trends. Importantly, the 4-bit models often match or outperform the full-precision baselines in calibration and uncertainty-based reliability metrics, suggesting that the reliability peak is not tied to a specific quantization method.
>
> > The core experiments remain centered on LLaMA-family models and QA-style evaluations
>
> Clarification of benchmark selection: We use standard benchmarks commonly adopted in prior LLM quantization work [1,2,3,4]. While previous works typically report the zero-shot performance to assess the effectiveness of quantization, we adopt these benchmarks for comparability, but significantly extend the analysis by examining (a) performance scalings and (b) reliability scalings, including uncertainty, calibration, and robustness to natural perturbations, in order to understand the underlying trends.
>
> Additional benchmarks are included in the appendix. We summarize all used benchmarks in Table 3. Please refer to Appendix D, where we provide inference bit-level scalings covering in-context learning tasks, instruction-following tasks, reasoning, and understanding tasks for LLaMA models, instruction-tuned LLaMA models, Qwen3 models, and OPT. Crucially, the observed trends are generally consistent across diverse model backbones.
>
> >The KL-divergence analysis helps describe behavioral shift, but it does not fully explain why 4-bit models should outperform 8-bit or 16-bit models
>
> We thank the reviewer for raising this critical point. We explain the reliability peak at 4-bit in the following: First, the KL-divergence analysis (Figure 6) explains why 4-bit precision outperforms 2 and 3-bit precisions, since the behavioral shift of 4-bit models to their full-precision counterparts is significantly smaller than that of 2- and 3-bit models. For the 4-bit vs 8-bit comparison, the key additional factor is model capacity under a fixed total-bit budget. A 4-bit model matched to an 8-bit model in total bits has roughly twice the number of parameters. Our results suggest that, in this regime, the benefit of increased model capacity can outweigh the benefit of higher weight precision: 4-bit models remain sufficiently close to their full-precision counterparts while benefiting from a larger parameter count. Below 4 bits, this trade-off changes: the behavioral shift induced by aggressive quantization becomes too large to be compensated by additional parameters. We update our manuscript to better clarify this point.

---

> > ### Author Response · Authors · 2026-05-22
> >
> > > the KL analysis is limited to three LLaMA models and does not cover the full range of architectures, tasks, perturbations, or quantization methods
> >
> > In the revised manuscript, we extend the KL divergence analysis beyond the LLaMA models. Specifically, we include LLaMA, OPT, and Qwen models, covering 2, 3, 4, and 8-bit precisions, and compare the behavioral shift of each quantized model relative to its corresponding full-precision base model on two different benchmarks. Please refer to Appendix I.
> >
> > The extended analysis supports the same qualitative conclusion: extreme low-bit quantization, especially 2-bit and 3-bit, induces a substantially larger behavioral shift, while 4-bit and 8-bit models remain considerably closer to their full-precision counterparts. This strengthens our interpretation that the degradation of extreme low-bit models is linked to larger distributional shifts, whereas the favorable behavior of 4-bit models reflects a trade-off between maintaining a small behavioral shift and benefiting from increased model capacity at a fixed total-bit budget. We note that conducting the KL analysis across the full range of architectures, tasks, perturbation settings, and quantization methods evaluated in the paper would be computationally prohibitive given the scale of our experimental setup. Nevertheless, the extended analysis in Appendix I provides representative evidence across multiple model backbones, quantization methods, bit-widths, and model sizes.
> >
> > > The authors should provide more statistical evidence to support that the 4-bit peak is statistically stable across datasets, perturbation types, random seeds, decoding settings, and quantization methods. Also, the current mechanism study remains too weak and empirical.
> >
> > To support the stability of our findings, we provide a comprehensive experimental evaluation across multiple dimensions: different model families (Section 4.2), benchmarks and tasks (Appendix D), compression techniques including six state-of-the-art PTQ methods as well as pruning and QAT variants (Section 4.2 and Appendix G), decoding-strategy ablations (Appendix F), and a broad set of natural, real-world perturbations. In the revised manuscript, we further strengthen this evidence by including all available quantized models in the scaling plots and fitting scaling curves per precision, rather than reporting only the best-performing model per bit-width. The resulting trends remain consistent, providing additional empirical support that the 4-bit regime is not an artifact of a specific quantization method, dataset, or plotting choice.
> >
> > Consistent with foundational scaling-law studies [5,6,7,8], our work is primarily empirical: we aim to uncover robust trends in model behavior under quantization and to identify patterns that can guide both theoretical understanding and practical model design. In this sense, our study helps constrain the design space of reliable and efficient LLMs by identifying total bit regimes that consistently provide favorable reliability–efficiency trade-offs.
> >
> >
> > [1] Frantar, Elias, et al. Gptq: Accurate post-training quantization for generative pre-trained transformers. (2022)
> >
> > [2] Lin, Ji, et al. Awq: Activation-aware weight quantization for on-device llm compression and acceleration. (2024)
> >
> > [3] Egiazarian, Vage, et al. Extreme compression of large language models via additive quantization. (2024)
> >
> > [4] Harma, Simla Burcu, et al. Effective interplay between sparsity and quantization: From theory to practice. (2024)
> >
> > [5] Kaplan et al., Scaling laws for neural language models. 2020
> >
> > [6] Chen et al., Q-resafe: Assessing Safety Risks and Quantization-aware Safety Patching for Quantized LLMs. 2025
> >
> > [7] Hoffmann, Jordan, et al. Training compute-optimal large language. 2022
> >
> > [8] Kumar, Tanishq, et al. Scaling laws for precision. 2024

---

### Decision · Action_Editor_ZSrz · 2026-06-12

**Recommendation:** Accept with minor revision

**Additional Comments:**

The reviewers reached a positive consensus after the revision. The authors have sufficiently addressed the main concerns regarding method selection effects, cross-quantization-method stability, and the scope of the scaling-law claim. For the final version, the authors should keep the wording cautious and consistently present the contribution as an empirical study of reliability scaling trends rather than as a universal predictive scaling law.

**Audience:**

Yes

**Audience Explanation:**

The paper is relevant to TMLR readers interested in LLM compression, quantization, calibration, uncertainty estimation, robustness, and trustworthy deployment.

**Claims And Evidence:**

Yes

**Claims Explanation:**

The paper provides sufficient empirical evidence for its main claim that reliability metrics for quantized LLMs can scale differently from standard performance metrics. The revision addresses the main concerns by including all available quantized models in the scaling plots, adding cross-method comparisons, and clarifying that the results should be interpreted as empirical scaling trends rather than universal predictive laws.